# AdaTS: Adaptive Time Series Representation Learning through Dynamic Contrasts

**Denizhan Kara**\*, **Tomoyoshi Kimura**\*, **Jinyang Li**\*‡, **Bowen He**†, **Yizhuo Chen**\*,
**Yigong Hu**\*, **Hongjue Zhao**\*, **Shengzhong Liu**†, **Tarek F. Abdelzaher**\*
\*University of Illinois Urbana-Champaign      †Shanghai Jiao Tong University
{kara4, tkimura4, jinyang7, yizhuoc, yigongh2, hongjue2, zaher}@illinois.edu
{hebowen1, shengzhong}@sjtu.edu.cn

## Abstract

Learning robust representations from unlabeled time series is crucial, and contrastive learning offers a promising avenue. However, existing contrastive learning approaches for time series often struggle to define meaningful similarities, tending to overlook inherent physical correlations and diverse, sequence-varying non-stationarity. This limits their representational quality and real-world adaptability. To address these limitations, we introduce AdaTS, a novel adaptive soft contrastive learning strategy. AdaTS offers a computationally efficient solution centered on dynamic instance-wise and temporal assignments that enhance time series representations by: *(i)* leveraging Time-Frequency Coherence to provide robust, physics-guided similarity measurements; *(ii)* preserving relative instance similarities through ordinal consistency learning; and *(iii)* adapting to sequence-specific non-stationarity with dynamic temporal assignments. AdaTS is designed as a pluggable module for standard contrastive frameworks, achieving accuracy improvements of up to 13.7% across diverse time series datasets and three state-of-the-art contrastive frameworks while enhancing robustness under label scarcity.

## 1  Introduction

Self-supervised learning (SSL) has emerged as a transformative paradigm for extracting meaningful representations from large-scale unlabeled data, achieving remarkable success in domains like computer vision [7, 25] and natural language processing [38]. Recently, SSL has garnered significant attention in time series (TS) analysis, particularly for Internet of Things (IoT) applications [37, 68, 29], which present unique challenges in data variability, non-stationarity, and labeling difficulty. As a popular SSL approach, contrastive learning (CL) structures semantically meaningful representations by distinguishing positive (*similar*) and negative (*non-similar*) sample pairs.

In TS data, representations from consecutive sampling periods are naturally correlated; nearby timestamps tend to have similar values because the underlying physical processes evolve smoothly (*e.g.,* movement patterns in activity recognition). With this property, existing works [68, 16] have proposed CL objectives by defining temporally close samples as similar and minimizing their geometric distances in the latent embedding space.

However, determining accurate similarity measures for TS CL faces multiple challenges: First, existing similarity metrics for TS often fail to effectively *capture inherent physical correlations*. These correlations arise from common underlying physical mechanisms, allowing signals from similar events or activities to be meaningfully compared even when separated in time, occurring under varied conditions, or originating from different instances (e.g., different individuals performing the same action or distinct machines of the same type in similar operational states). Many standard TS similarity

---

‡Corresponding author.

39th Conference on Neural Information Processing Systems (NeurIPS 2025).

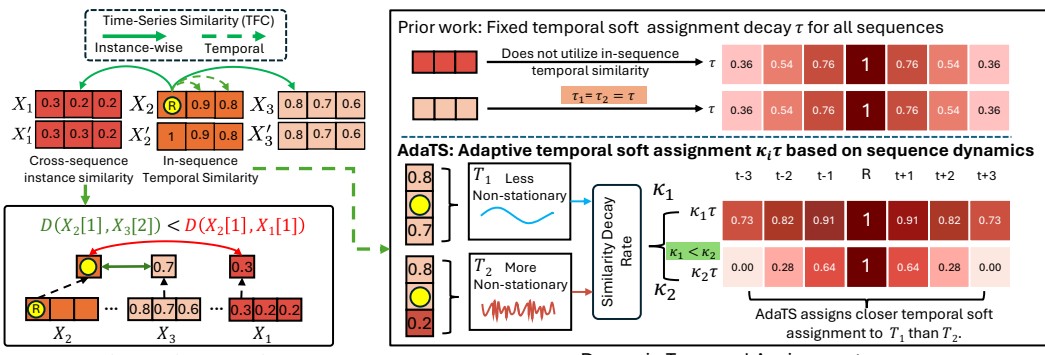

Figure 1: Overview of AdaTS framework. Best viewed in color.

methods (e.g., cosine, Euclidean) overlook underlying physical signal properties and are susceptible to noise [6, 31]. As a widely used TS distance metric, Dynamic Time Warping (DTW) [49, 15] enables temporal alignment between sequences but is computationally expensive and impractical for iterative training on large-scale datasets. Moreover, DTW is sensitive to noise and pathological alignments in high-resolution data, limiting its general applicability in extensive IoT applications [52, 47]. Prior works like FastDTW [50] have attempted to optimize DTW's computational efficiency but at the cost of accuracy degradation on high-frequency data.

Second, defining consistent similarity relations must account for *non-stationary TS characteristics*. Real-world signals often exhibit substantial variability not only within a single sequence but also *across different sequences*, where statistical properties and temporal dynamics evolve due to factors like motion patterns, environmental changes, or varying system states [35, 11]. Existing CL frameworks often miss this inter-sequence variability, either by assigning all negative pairs or temporal relationships uniformly [63, 17, 68], or by adopting the same decay functions for temporal similarity (e.g., a single sigmoid function for temporal similarity decay for all sequences) [34]. Such static approaches fail to adapt to the varying rates of change and temporal unpredictability inherent in different sequences even within the same dataset.

To address these challenges, we propose AdaTS, an adaptive soft contrastive learning strategy. AdaTS is a flexible, pluggable module designed to augment existing CL frameworks by dynamically modeling both instance-wise physical similarities and sequence-specific temporal characteristics with minimal computational overhead. Figure 1 provides an overview of AdaTS: a time-series similarity metric (green arrows) establishes temporal and cross-sequence relations. The similarities across different sequence samples guide an ordinal consistency loss for instance-wise learning of soft similarity relations across sequences, while similarities within samples of each sequence introduce a dynamic temporal assignment mechanism (right panel) that adapts temporal similarity weights based on each sequence's unique temporal variability. Its core contributions are:

1. **Physics-Guided Time-Frequency Similarity:** While AdaTS's components can integrate various TS similarity metrics, we leverage Time-Frequency Coherence (TFC) [70] as a primary similarity measure. TFC robustly quantifies similarity in the frequency domain by analyzing harmonic structures. We apply it within our framework, which provides key advantages such as computational efficiency and robustness for TS similarity relations (see Section 2.2).

2. **Instance-wise Ordinal Consistency Learning:** Leveraging TS sample similarities across sequences, we propose an ordinal consistency loss (Section 2.3). Instead of enforcing absolute similarity values in time series contrasting, which can be noisy and context-dependent, this loss preserves the *relative* ordering of physical similarities between samples. This self-supervised approach differs from methods using progressive augmentation intensities to learn ordinal relationships [30] or requiring explicit ordinal labels [2] by deriving relations directly from signal properties.

3. **Dynamic Temporal Similarity Assignment:** Leveraging temporal similarities within TS sequences, we introduce a dynamic temporal assignment mechanism (Section 2.4) that adapts to the varying dynamics of different sequences. This mechanism quantifies the average adjacent dissimilarity of each sequence to dynamically adjust its temporal similarity weights within contrastive learning. This approach reflects the varying temporal dynamics and non-stationarities inherent to each sequence, allowing a more robust and accurate TS similarity relation.

**Algorithm 1** TIMEFREQUENCYCOHERENCE

---

**Require:** Dataset $\mathcal{X} = \{\mathbf{X}_1, ..., \mathbf{X}_N\}, \mathbf{X}_i \in \mathbb{R}^{T \times L \times C}$      $\triangleright N$ sequences with $M = T * N$ samples $x$
1:  *% Phase 1 - Compute STFT and auto-spectral densities*
2:  **for** $x_i$ to $\mathcal{X}$ **do**
3:     $x_i(t, f) \leftarrow \text{STFT}(\mathbf{x}_i)$                 $\triangleright$ Time-frequency representation
4:     $S_{ii}(t, f) \leftarrow |x_i(t, f)|^2$               $\triangleright$ Auto-spectral density
5:  **end for**
6:  *% Phase 2 - Compute pairwise coherence across all samples*
7:  $C \leftarrow \mathbb{R}^{M \times M}$                          $\triangleright$ Initialize coherence matrix
8:  **for** $i = 1$ to $M$ **do**
9:     **for** $j = 1$ to $M$ **do**
10:       $S_{ij}(t, f) \leftarrow x_i(t, f) \cdot x_j(t, f)^*$        $\triangleright$ Cross-spectral density
11:       $C_{ij} \leftarrow \frac{1}{L \cdot F} \sum_{t,f} \frac{|S_{ij}(t,f)|^2}{S_{ii}(t,f) \cdot S_{jj}(t,f)}$       $\triangleright$ Average TFC
12:     **end for**
13:  **end for**
14:  **return** $C$                      $\triangleright M \times M$ pairwise coherence similarity

---

We extensively evaluate AdaTS across six TS datasets, demonstrating its ability to augment existing methods and enhance performance. Specifically, AdaTS improves average CL accuracy by 7.3% (up to 13.7%), enhances robustness to dynamic variations, and achieves superior performance at low label rates. Our results show that AdaTS effectively incorporates underlying physical and temporal correlations and adapts to the varying sequence dynamics common in real-world TS applications.

## 2   AdaTS Framework

This section presents the AdaTS framework. We first define the problem setting and then detail its three complementary components: *(i)* Time-Frequency Coherence (TFC), *(ii)* Ordinal Consistency Learning, and *(iii)* Dynamic Temporal Assignment.

### 2.1   Problem Definition

We address the problem of learning a nonlinear embedding function $f_\theta$ given a dataset $\mathcal{X} = \{\mathbf{X}_1, \ldots, \mathbf{X}_N\}$ of $N$ time series sequences. Each sequence $\mathbf{X}_i \in \mathbb{R}^{T \times L \times C}$ comprises $T$ samples, $\{\mathbf{x}_{i,1}, \ldots, \mathbf{x}_{i,T}\}$, where each sample $\mathbf{x}_{i,t} \in \mathbb{R}^{L \times C}$ is a fixed-length window of $L$ sensor readings from $C$ channels. The objective is to learn $f_\theta : \mathbb{R}^{L \times C} \to \mathbb{R}^H$, mapping each sample $\mathbf{x}_{i,t}$ to an $H$-dimensional representation $r_{i,t} = f_\theta(\mathbf{x}_{i,t})$.

### 2.2   Time-Frequency Coherence

A critical challenge in time series similarity analysis is handling non-stationarity, where a time series' statistical properties (*e.g.,* mean or variance) evolve over time due to structural shifts or external influences. Such variations manifest as dynamic trends, changing noise levels, or shifts in frequency content [11]. To efficiently handle such dynamic time series properties, we leverage Time-Frequency Coherence (TFC) as the *similarity measure* that enforces spectral consistency. TFC ensures robust, efficient signal similarity across both time and frequency domains. By capturing harmonic alignments with FFT-based operations, TFC overcomes the computational overhead of DTW and remains resilient to noise. TFC captures spectral relationships that often correlate with underlying physical processes (*e.g.,* harmonic structures in vehicle engines or periodic patterns in human movements), providing a meaningful way to compare signals across varying conditions without restricting AdaTS from using other similarity metrics.

TFC addresses the limitations of conventional time series similarity metrics like DTW in three key ways: (i) It provides computational efficiency through FFT-based operations, making it suitable for large datasets with high sample rates [19, 24], (ii) It captures spectral relationships without introducing non-linear temporal deformations that can distort signal characteristics [50, 52], (iii) It demonstrates robustness to non-stationarity and noise [24], and generalizes to different domains [58, 40, 46].

We detail TFC in Algorithm 1, consisting of two main phases. In Phase 1, we compute the Short-Time Fourier Transform (STFT) for each sample $\mathbf{x}_i$ to obtain their time-frequency representation $x_i(t, f)$. We then calculate the auto-spectral density $S_{ii}(t, f)$ for each sample, which represents its

---

**Algorithm 2** ORDINAL CONSISTENCY LOSS

---

**Require:** Dataset embeddings $\mathcal{R} = \{r_1, ..., r_M\}$, $r_i \in \mathbb{R}^H$; Similarity matrix $\mathbf{D} \in \mathbb{R}^{M \times M}$; margin $\delta$

1: $\mathcal{T} \leftarrow \{(i,j,k) \mid D(i,j) < D(i,k), \forall i,j,k \in [1, M]\}$ ▷ Valid triplets with semantic ordering
2: $d(i,j) \leftarrow \|r_i - r_j\|_2$ ▷ Euclidean distance
3: $\mathcal{L}_{\text{OC}} \leftarrow \frac{1}{|\mathcal{T}|} \sum_{(i,j,k) \in \mathcal{T}} \max\{0, d(i,j) - d(i,k) + \delta\}$ ▷ Enforce $d(i,j) + \delta \leq d(i,k)$
4: **return** $\mathcal{L}_{\text{OC}}$ ▷ Average loss over all valid triplets

---

power spectral density. In Phase 2, we compute the cross-spectral density $S_{ij}(t, f)$ between each pair of samples $i$ and $j$, which captures the similarity between two samples from time-frequency representations. We also calculate the time-frequency coherence $C_{ij}(t, f)$ between samples $i$ and $j$ by normalizing the cross-spectral density with the auto-spectral densities. We use magnitude-squared coherence following the standard TFC definitions [19, 70, 24], which intentionally discards sign information to quantify linear correlation strength. $C_{ij}(t, f)$ represents the similarity between the frequency components of samples at time $t$ and frequency $f$. Finally, we average the coherence values across all time-frequency bins to calculate the time-frequency coherence $C_{ij}$.

TFC provides a semantically meaningful measure by capturing harmonic structures without introducing non-linear distortions. It is computationally efficient, sample-rate agnostic, and well-suited for large-scale datasets with varying frequency resolutions. Additionally, TFC offers a straightforward measure to capture the change in relevant dynamics within time series sequences. We use these properties to enhance the ordinal consistency learning and dynamic temporal assignment of AdaTS.

## 2.3 Ordinal Consistency Learning

Traditional CL approaches like InfoNCE [42] contrast all instances within a batch via hard negatives. However, directly applying hard negative sampling to TS data can be problematic, as it overlooks the inherent temporal sample similarities. To address this, prior works [34] use time series similarity metrics (*e.g.,* DTW) to structure the representation space by enforcing semantic similarity to reflect the data-level similarity. However, this assumption does not always hold. First, data-level similarity metrics are inherently context-dependent, whose effectiveness varies based on the characteristics of the dataset and the specific domain. These metrics could introduce biases that can limit generalization across different types of time series data. Second, due to the non-stationary nature of time series signals and the prevalence of noise, these metrics may fail to accurately capture the underlying correlations between events and true physical phenomena. Consequently, enforcing semantic similarity to match TS similarity metrics can corrupt the representations and struggle to generalize across diverse TS datasets. For example, vehicle signals recorded in real-world environments often contain substantial noise, which can distort metric-based similarities. Under such scenarios, using distorted values to enforce strict semantic alignment could severely corrupt the representations. To mitigate this, we propose an ordinal consistency loss to restrict the relative ordering of similarities, which remains mostly consistent within the same data context and, therefore, more generalizable.

Algorithm 2 details the ordinal consistency loss. Given sample embeddings $\mathcal{R} = \{r_1, ..., r_M\}$ and their similarity matrix $\mathbf{D} \in \mathbb{R}^{M \times M}$, valid sample triplets $\mathcal{T}$ that preserve relative similarity ordering are generated. Then, Euclidean distances $d(i, j)$ are computed between embeddings $r_i$ and $r_j$. The ordinal consistency loss $\mathcal{L}_{\text{OC}}$ is then calculated as the average margin ranking loss over valid triplets $\mathcal{T}$, ensuring the positive-to-reference distance is less than the negative-to-reference distance. Triplet generation is implemented efficiently with vectorized tensor operations for minimal overhead.

The ordinal loss ensures that embedding distance orders align with the physical characteristics of samples. Unlike methods relying on absolute similarity values (which can be unreliable due to noise, context-dependency, and varying physical factors like motion or environmental conditions), our ordinal consistency loss focuses solely on the relative ordering between sample pairs. This approach is well-suited for physical signals, as relative distance orders provide more stable, reliable indicators of semantic similarity across diverse conditions than absolute similarity measurements.

## 2.4 Dynamic Temporal Assignment

Compared to prior works that employ static temporal similarity assignments [34, 63], we observe that time series sequences within the same dataset can exhibit varying non-stationarity and different rates of change. While tuning a "sharpness" parameter $\tau_T$ for a decay function can help, this one-size-fits-all parameter still forces the model to globally apply the same setting on faster-changing and

---

**Algorithm 3** DYNAMIC TEMPORAL CONTRASTIVE LOSS

---

**Require:** Dataset $\mathcal{X} = \{\mathbf{X}_1, ..., \mathbf{X}_N\}$ where $\mathbf{X}_i \in \mathbb{R}^{T \times L \times C}$; similarity function $\mathcal{D}$; embedding function $f_\theta$.
1: Let $r_{i,t} = r_{i,t+2T}$ and $\tilde{r}_{i,t} = r_{i,t+T}$ be embeddings from two augmentations of $\mathbf{X}_i$ at timestamp $t$.
2: *% Phase 1 - Compute pairwise similarity for each sequence*
3: **for** $n = 1$ to $N$ **do**
4:     $\mathbf{D}_n^T \in \mathbb{R}^{T \times T} \leftarrow \mathcal{D}(\mathbf{X}_n)$
5: **end for**
6: *% Phase 2 - Compute similarity decay rate for each sequence*
7: **for** $n = 1$ to $N$ **do**
8:     $\kappa^n \leftarrow -\frac{1}{T-1} \sum_{t=0}^{T-2} \log(\mathbf{D}_n^T[t, t+1] + \epsilon)$                      $\triangleright$ $X_n$ similarity decay rate
9: **end for**
10: $\kappa^n \leftarrow \text{BatchNorm}(\kappa^n)$
11: *% Phase 3 - Temporal soft assignment with dynamic decay*
12: **for** $n = 1$ to $N$ **do**
13:     $w_T^n(t, t') = 2\,\sigma(-(\kappa^n \tau_T) \cdot |t - t'|)$              $\triangleright$ Soft assignment for timestamps $t$ and $t'$
14:     $p_T(i, (t, t')) \leftarrow \dfrac{\exp\left(r_{i,t} \circ r_{i,t'}\right)}{\sum\limits_{s=1,\, s \neq t}^{2T} \exp\left(r_{i,t} \circ r_{i,s}\right)}$                         $\triangleright$ Softmax
15:     $\mathcal{L}^+(i, t) \leftarrow -\log\left(p_T(i, t+T)\right)$                $\triangleright$ Compute CL positive pairs
16:     $\mathcal{L}^-(i, t) \leftarrow -\sum\limits_{s \neq \{t, t+T\}} w_T^n(t, s) \log p_T(i, s)$       $\triangleright$ Compute negative pairs
17:     $\ell_T^{(i,t)} = \mathcal{L}^+(i, t) + \mathcal{L}^-(i, t)$                   $\triangleright$ Loss for each timestamp
18: **end for**
19: **return** $\mathcal{L}_T = \frac{1}{4NT} \sum_{n=1}^{2N} \sum_{t=1}^{2T} \ell_T^{(i,t)}$

---

slower-changing sequences alike, often resulting in suboptimal modeling of rapid signal variations. Instead, we propose dynamically assigning a varying degree of temporal similarity weighting to each sequence based on its intrinsic temporal characteristics, enabling the framework to accommodate different levels of dynamics more effectively. This adaptive strategy can better model the temporal evolution of signals with varying rates of change.

Inspired by Schreiber's method [51] to exploit information in time series similarities, we propose calculating a statistic $\kappa^n$ for each sequence $\mathbf{X}_n$ to detect physically relevant temporal differences and changes via dissimilarity of adjacent samples to adapt the temporal similarity assignment. The intuition behind $\kappa^n$ is to measure the overall temporal unpredictability within a sequence. A high $\kappa^n$ indicates that, on average, adjacent time steps have low similarity (high dissimilarity), suggesting the sequence exhibits rapidly changing dynamics or is generally less predictable. Conversely, a low $\kappa^n$ indicates high average similarity between adjacent steps, suggesting smoother transitions or more predictable temporal structures. This statistic allows for dynamic adaptation of temporal similarity assignments for TS sequences by identifying sequences with rapid changes or stable regions.

Algorithm 3 summarizes the dynamic temporal assignment process. In Phase 1, given the dataset $\mathcal{X} = \{\mathbf{X}_1, ..., \mathbf{X}_N\}$ and a similarity function $\mathcal{D}$ (e.g., TFC), we first compute the intra-sequence pairwise similarity $\mathbf{D}_n^T$ for each sequence $\mathbf{X}_n$. $\mathbf{D}_n^T$ is a $T \times T$ matrix where each element $d_{t,t'}$ represents the similarity between samples at timestamps $t$ and $t'$ in sequence $\mathbf{X}_n$. In Phase 2, we calculate the average adjacent dissimilarity statistic $\kappa^n$ for each sequence $\mathbf{X}_n$ to measure its inherent temporal unpredictability. For each sequence, we compute the negative logarithm of the similarity between adjacent timestamps. This transformation maps similarity values (typically in $[0, 1]$) to a positive dissimilarity measure: low similarity (near 0) yields a large positive value, while high similarity (near 1) results in a small positive value. Averaging these transformed values yields $\kappa^n$, representing the sequence's average dissimilarity per transition. Using the negative logarithm makes $\kappa^n$ more sensitive to instances of very low adjacent similarity compared to a direct average. A few steps with near-zero similarity will dramatically increase $\kappa^n$, which is desirable to detect sharp drops in coherence and significant changes in dynamics. Normalization across the batch is then applied to ensure the statistic $\kappa^n$ is comparable across different sequences.

In Phase 3, we calculate the temporal soft assignment weights $w_T^n(t, t')$ for each pair of timestamps $(t, t')$ through a sigmoid function. Our approach extends SoftCLT [34] by introducing a sequence-specific dynamic decay rate $\kappa^n \cdot \tau_T$. Here, the base rate $\tau_T$ is modulated by each sequence's average adjacent dissimilarity statistic $\kappa^n$. This dynamic decay rate allows the model to adapt the temporal similarity assignment based on the sequence's inherent unpredictability. Sequences with higher $\kappa^n$

Table 1: Finetune results on different datasets. We mark the **best** results.

| Metric | ACIDS Acc | ACIDS F1 | MOD Acc | MOD F1 | PAMAP2 Acc | PAMAP2 F1 | RWHAR Acc | RWHAR F1 | Average Acc | Average F1 |
|---|---|---|---|---|---|---|---|---|---|---|
| SW-T [39] | 0.9187 | 0.7829 | 0.8935 | 0.8919 | 0.8462 | 0.8148 | 0.9031 | 0.9018 | 0.8904 | 0.8479 |
| TST [69] | 0.6785 | 0.5010 | 0.5820 | 0.5371 | 0.7469 | 0.6692 | 0.7928 | 0.6757 | 0.7001 | 0.5958 |
| MAE [25] | 0.8521 | 0.6908 | 0.7817 | 0.7793 | 0.7382 | 0.6999 | 0.8638 | 0.8700 | 0.8089 | 0.7600 |
| AudioMAE [27] | 0.7845 | 0.6120 | 0.7274 | 0.7249 | 0.7808 | 0.7478 | 0.8163 | 0.7437 | 0.7773 | 0.7071 |
| CAVMAE [22] | 0.7995 | 0.6711 | 0.5432 | 0.5266 | 0.7995 | 0.6711 | 0.9113 | 0.9153 | 0.7634 | 0.6960 |
| CMC [54] | 0.7836 | 0.6452 | 0.9049 | 0.9023 | 0.7571 | 0.7223 | 0.8211 | 0.8384 | 0.8167 | 0.7771 |
| TNC [55] | 0.8352 | 0.7372 | 0.8533 | 0.8539 | 0.8013 | 0.7506 | 0.8817 | 0.8784 | 0.8429 | 0.8050 |
| Cosmo [43] | 0.8776 | 0.7298 | 0.3228 | 0.3241 | 0.7910 | 0.7469 | 0.8529 | 0.7968 | 0.7111 | 0.6494 |
| SimCLR [7] | 0.5658 | 0.4879 | 0.7535 | 0.7434 | 0.7346 | 0.6635 | 0.7830 | 0.7181 | 0.7092 | 0.6532 |
| TF-C [71] | 0.7863 | 0.6448 | 0.5787 | 0.5712 | 0.6593 | 0.6058 | 0.7998 | 0.7049 | 0.7060 | 0.6317 |
| MF-CLR [14] | 0.8343 | 0.6587 | 0.8058 | 0.8042 | 0.7445 | 0.7045 | 0.7940 | 0.7954 | 0.7947 | 0.7407 |
| Informer [72] | 0.9470 | 0.8455 | 0.8972 | 0.8961 | 0.8746 | 0.8660 | 0.9313 | 0.9353 | 0.9125 | 0.8857 |
| LIMU-BERT [65] | 0.5556 | 0.3712 | 0.4297 | 0.3970 | 0.7781 | 0.7554 | 0.8120 | 0.7508 | 0.6439 | 0.5686 |
| TS2Vec [68] | 0.7703 | 0.6312 | 0.7380 | 0.7350 | 0.6069 | 0.5280 | 0.6117 | 0.6019 | 0.6817 | 0.6240 |
| TS2Vec + SoftCLT | 0.7753 | 0.6679 | 0.7495 | 0.7509 | 0.7047 | 0.6272 | 0.8226 | 0.7909 | 0.7630 | 0.7092 |
| TS2Vec + AdaTS | 0.8379 | 0.7211 | 0.7810 | 0.7825 | 0.7772 | 0.7266 | 0.8794 | 0.8877 | 0.8189 | 0.7795 |
| TS-TCC [16] | 0.8758 | 0.7400 | 0.7709 | 0.7744 | 0.7871 | 0.7107 | 0.8684 | 0.8227 | 0.8256 | 0.7620 |
| TS-TCC + SoftCLT | 0.8925 | 0.7784 | 0.7894 | 0.7865 | 0.8034 | 0.7323 | 0.8875 | 0.8680 | 0.8432 | 0.7913 |
| TS-TCC + AdaTS | 0.9155 | 0.8027 | 0.8135 | 0.8120 | 0.8257 | 0.7764 | 0.8984 | 0.8712 | 0.8633 | 0.8156 |
| FOCAL [37] | 0.9347 | 0.8272 | 0.9548 | 0.9540 | 0.8438 | 0.8243 | 0.9261 | 0.9327 | 0.9148 | 0.8846 |
| FOCAL + SoftCLT | 0.9425 | 0.7991 | 0.9564 | 0.9554 | 0.8498 | 0.8298 | 0.9383 | 0.9433 | 0.9218 | 0.8819 |
| FOCAL + AdaTS | **0.9571** | **0.8480** | **0.9705** | **0.9700** | **0.8648** | **0.8560** | **0.9527** | **0.9568** | **0.9363** | **0.9077** |

(more unpredictable, lower adjacent similarity) will experience a faster decay of temporal similarity weights, while smoother sequences (lower $\kappa^n$) will have a slower decay. Since contrastive loss can be interpreted as the cross-entropy loss [33, 34], we define a softmax probability $p_T(i, (t, t'))$ of the relative similarity against all other pairs when computing loss. Following similar convention, we use simplified indexing e.g., $r_{i,t+T}, r_{i,t+2T}$ to refer to the concatenated augmented views of the same sample. Lastly, we compute the weighted contrastive loss for each sequence where the positive pairs are timestamp pairs separated by a time lag of $T$, and negative pairs consider other timestamps in the sequence weighted by the soft assignment weights $w_T^n(t, t')$.

The final AdaTS contrastive loss combines ordinal consistency $\mathcal{L}_{OC}$ and temporal contrastive $\mathcal{L}_T$ components, to be combined with the original loss from the base SSL framework:

$$\mathcal{L}_{AdaTS} = \lambda_{oc}\mathcal{L}_{OC} + \lambda_t\mathcal{L}_T, \tag{1}$$

where $\lambda_{oc}, \lambda_t \in [0, 1]$ control the contribution of each loss.

## 3 Evaluation

We present our experimental setup and extensive evaluations to demonstrate AdaTS's performance and efficiency. We further ablate AdaTS to understand the contributions of its components.

### 3.1 Experimental Setup

**Datasets.** We extensively evaluate AdaTS across two IoT application domains using four datasets: Human Activity Recognition (PAMAP2 [48], RWHAR [53]) and Moving Object Detection (MOD [37], ACIDS [62]). We segment time-series data into samples of fixed-size windows, with lengths determined by each dataset's temporal characteristics. For multi-modal data, AdaTS processes each modality independently to accommodate varying sampling rates and resolutions, aligning with our channel-wise soft assignment mechanism. We partition each dataset into training, validation, and test sets using an 8:1:1 ratio. Detailed dataset descriptions and configurations are provided in Appendix A.

**Baselines and Backbone Model.** We compare AdaTS with 12 SOTA baselines spanning contrastive [54, 43, 7], masked reconstruction [25, 65, 27, 22], Transformer-based [69, 39], and temporal [68, 16, 55, 37] SSL frameworks. Details are provided in Appendix B. We employ SWIN-Transformer (SW-T) [39] as the backbone, which computes local attention within sliding windows across input spectrogram patches. For fine-tuning, we append a linear classification layer to the pretrained representations. Training details and configurations are available in Appendix C and G.

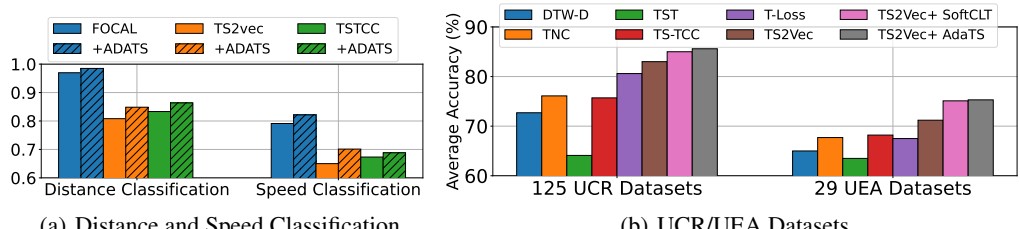

(a) Distance and Speed Classification                    (b) UCR/UEA Datasets

Figure 2: Classification performance across MOD tasks (left) and UCR/UEA datasets (right).

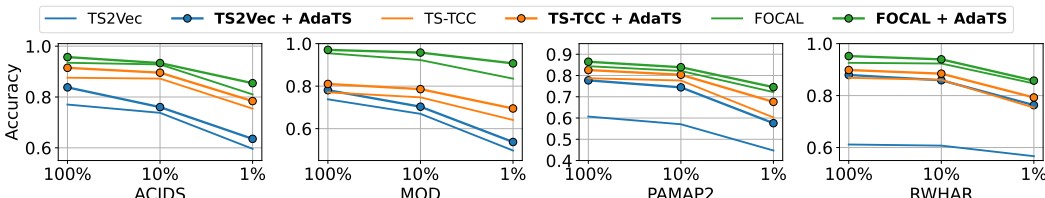

Figure 3: Accuracy with or without AdaTS across different label ratios (100%, 10%, 1%).

## 3.2 Evaluation Results

**Fine-tuning Results.** Table 1 compares AdaTS's performance with other self-supervised learning frameworks across different datasets. Results show that AdaTS improves performance across all three contrastive frameworks. When integrated with TS2Vec, AdaTS achieves the largest average improvement of 13.7%, while integration with FOCAL yields the best absolute performance with a 2.1% accuracy gain.

The HAR datasets (PAMAP2 and RWHAR) benefit most from AdaTS due to their relatively lower complexity and bandwidth compared to vehicle classification tasks [1, 62], enabling better capture and differentiation of similarity relations. In contrast, the Vehicle Classification datasets (MOD and ACIDS) show less improvement due to their high dynamics and complex non-stationarities from environmental conditions, which challenge instance-wise similarity capture. However, AdaTS's dynamic temporal assignment mechanism and frequency-aware similarity enable more robust CL representations compared to existing baselines and SoftCLT, leading to better performance. Moreover, we observe that FOCAL and TS2Vec benefit relatively more from AdaTS compared to TSTCC. This is because the TS-TCC framework is already built upon leveraging the temporal correlation of sensing signals through temporal contrasting views. Hence, the additional temporal contrastive loss in AdaTS does not provide as much additional information as it does for FOCAL and TS2Vec, which are more focused on CL objectives through orthogonality relations and hierarchical feature learning.

We evaluate AdaTS-augmented CL baselines on additional downstream tasks of distance and speed classification on the MOD dataset. Figure 2(a) shows AdaTS consistently improves embedding quality across all baselines, leading to better downstream performance. Notably, we observe larger gains on speed classification tasks, which require more fine-grained temporal pattern recognition based on sequence dynamics. This demonstrates that AdaTS's dynamic temporal assignment mechanism and ordinal consistency effectively capture varying motion patterns, enabling better speed differentiation in non-stationary sequences. Qualitative assessment through t-SNE visualizations (Appendix D.1) further confirms that AdaTS produces more distinct and well-separated clusters across all datasets, particularly improving separation for complex, high-frequency dynamics in ACIDS and MOD.

We further evaluate AdaTS on TS classification tasks using 125 UCR archive univariate datasets [12] and 29 UEA archive multivariate datasets [3]. We implement AdaTS with TS2Vec using the SoftCLT evaluation code [34]. Figure 2(b) shows that AdaTS consistently outperforms the baselines across both UCR and UEA datasets. The performance gains over SoftCLT are smaller compared to IoT datasets, since UCR and UEA datasets exhibit more stationary patterns with less noise and dynamics. Consequently, the dynamic temporal assignments provide less additional information compared to IoT datasets. Nevertheless, AdaTS's superior performance demonstrates its robustness and adaptability across different datasets.

**Label Efficiency in Fine-tuning.** Next, we evaluate AdaTS with limited fine-tuning labels. In Figure 3, we present the accuracy curves of different CL baselines with and without AdaTS using 100%, 10%, and 1% of training labels. We observe that CL frameworks with AdaTS consistently

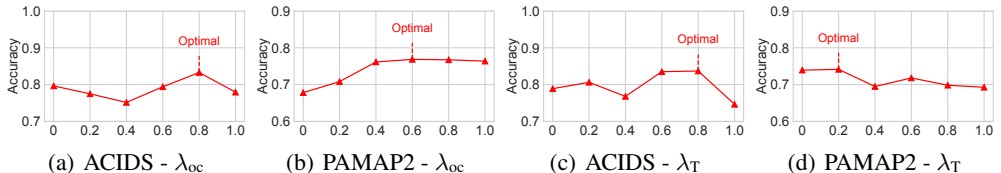

| (a) ACIDS - $\lambda_{\text{oc}}$ | (b) PAMAP2 - $\lambda_{\text{oc}}$ | (c) ACIDS - $\lambda_{\text{T}}$ | (d) PAMAP2 - $\lambda_{\text{T}}$ |

Figure 4: Ordinal consistency ($\lambda_{\text{oc}}$) and dynamic assignment ($\lambda_t$) hyperparameter analysis (TS2Vec).

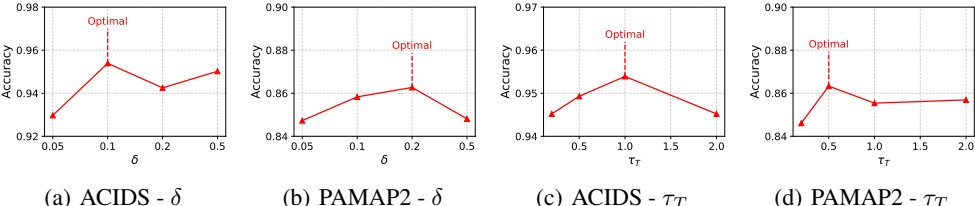

| (a) ACIDS - $\delta$ | (b) PAMAP2 - $\delta$ | (c) ACIDS - $\tau_T$ | (d) PAMAP2 - $\tau_T$ |

Figure 5: Sensitivity analysis for ordinal consistency margin ($\delta$) and dynamic temporal assignment temperature ($\tau_T$) (FOCAL).

outperform their counterparts across all label ratios, validating AdaTS's capability in improving the performance of SSL frameworks even with limited labels. Notably, there are consistent performance improvements across all label ratios, with the largest gains usually observed at 1% labels. This is due to the increased difficulty in learning meaningful representations with fewer labels, where AdaTS's instance-wise temporal relationships and dynamic assignment mechanisms provide more granular information to guide the embedding space.

**Robustness Evaluations.** We evaluate AdaTS's robustness under challenging conditions including noise corruption and extreme class imbalance. Comprehensive experiments with Gaussian corruption at multiple levels ($\sigma = 0.1$ to $1.0$) demonstrate that AdaTS consistently outperforms baselines, with F1 improvements up to +7.31% on ACIDS under extreme noise (detailed in Appendix D.3). TFC's spectral coherence analysis filters noise while preserving signal correlations, and ordinal consistency maintains relative similarity relationships when absolute metrics fail. Beyond noise robustness, we also evaluate performance under extreme class imbalance. While ordinal consistency is robust to moderate data variations, extreme class imbalance can cause embedding space shrinkage when similar instances are over-represented. We address this by applying distance regularization following the VICReg anti-collapse framework [4]. Controlled oversampling experiments (Appendix D.4) show that distance regularization recovers 52% of performance loss under extreme $10\times$ oversampling conditions, effectively maintaining embedding space diversity. We recommend applying distance regularization when dataset imbalance exceeds $5\times$ oversampling ratios.

**Sensitivity Analysis.** We analyze the sensitivity of AdaTS to key hyperparameters: ordinal consistency weight $\lambda_{\text{oc}}$, dynamic temporal assignment weight $\lambda_t$, ordinal consistency margin $\delta$ (Algorithm 2), and dynamic temporal assignment temperature $\tau_T$ (used in conjunction with Algorithm 3).

Figure 4 presents the analysis for $\lambda_{\text{oc}}$ and $\lambda_t$ on ACIDS and PAMAP2 datasets. Increasing $\lambda_{\text{oc}}$ prioritizes instance-wise relationships across sequences. We observe that both ACIDS and PAMAP2 benefit from higher $\lambda_{\text{oc}}$ values, with PAMAP2 showing more consistent improvements due to its lower sample complexity and dynamics. This allows for more effective capture of granular similarities through ordinal consistency. On the other hand, increasing $\lambda_t$ enhances the dynamic temporal assignment mechanism, which particularly benefits ACIDS due to its complex dynamics and higher non-stationarity. The dynamic assignment helps capture varying temporal relationships in sequences with diverse motion patterns and environmental conditions.

Figure 5 presents sensitivity analysis for the ordinal consistency margin $\delta$ and dynamic temporal assignment temperature $\tau_T$. The ordinal consistency margin $\delta$ controls the separation enforced between positive and negative pairs in the ordinal triplet loss. For ACIDS, with highly dynamic, high-frequency signals, smaller $\delta$ values (e.g., 0.1-0.2) preserve fine-grained similarity distinctions crucial for differentiating similar events. For PAMAP2, with smoother, lower-frequency signals, a slightly larger $\delta$ (e.g., 0.2) helps enforce clearer separation in the embedding space. Overall, moderate values ($\delta \in [0.1, 0.2]$) yield robust performance. The dynamic temporal assignment temperature $\tau_T$ modulates how strictly the temporal similarity decay influences the temporal contrastive loss. For ACIDS, characterized by continuous vehicle motion, moderate to higher $\tau_T$ values (e.g., 0.5-1.0)

Table 2: Compute Pretrain Overhead Comparison.

| Model | TS2Vec | + AdaTS | TS-TCC | + AdaTS | FOCAL | + AdaTS |
|---|---|---|---|---|---|---|
| Size (MB) | 2685 | 2693 | 2787 | 2797 | 2663 | 2670 |
| Time (s) | 0.231 | 0.243 | 0.257 | 0.271 | 0.232 | 0.245 |

Table 3: Distance Function Overhead.

| Dist Func | Euclidean | Cosine | DTW | TFC |
|---|---|---|---|---|
| Time (s) | 0.206 | 0.245 | 10.895 | 0.245 |
| Mem. (MB) | 7.8 | 7.8 | 19.6 | 7.8 |

Table 4: AdaTS component ablation results.

| Backbone | Components | PAMAP2 | | ACIDS | |
|---|---|---|---|---|---|
| | | Acc | F1 | Acc | F1 |
| TS2Vec | Baseline | 0.6069 | 0.5280 | 0.7703 | 0.6312 |
| | noOrd | 0.6782 | 0.6101 | 0.7964 | 0.6992 |
| | noTemp | 0.7393 | 0.6700 | 0.7890 | 0.6857 |
| | wStaticTemp | 0.7527 | 0.7006 | 0.8153 | 0.7073 |
| | AdaTS | 0.7772 | 0.7266 | 0.8379 | 0.7211 |
| FOCAL | Baseline | 0.8438 | 0.8243 | 0.9347 | 0.8272 |
| | noOrd | 0.8454 | 0.8309 | 0.9415 | 0.8421 |
| | noTemp | 0.8537 | 0.8380 | 0.9500 | 0.8361 |
| | wStaticTemp | 0.8551 | 0.8417 | 0.9464 | 0.8326 |
| | AdaTS | **0.8648** | **0.8560** | **0.9571** | **0.8480** |

Table 5: AdaTS similarity metric results. Best results in **bold**, second-best underlined.

| Backbone | Sim. Function | PAMAP2 | | ACIDS | |
|---|---|---|---|---|---|
| | | Acc | F1 | Acc | F1 |
| TS2Vec | TFC | **0.7772** | **0.7266** | **0.8379** | **0.7211** |
| | DTW | 0.7575 | 0.7106 | 0.7837 | 0.6740 |
| | Euclidean | 0.7469 | 0.6853 | 0.7998 | 0.6644 |
| | Cosine | 0.7244 | 0.6861 | 0.8066 | 0.6904 |
| FOCAL | TFC | **0.8648** | **0.8560** | **0.9571** | **0.8480** |
| | DTW | 0.8482 | 0.8408 | 0.9443 | 0.8381 |
| | Euclidean | 0.8592 | 0.8468 | 0.9324 | 0.8289 |
| | Cosine | 0.8403 | 0.8339 | 0.9475 | 0.8419 |

perform well. For PAMAP2, with more abrupt motion changes, lower to moderate $\tau_T$ values (e.g., 0.5) are effective. The robust performance across $\tau_T \in [0.5, 1.0]$ indicates that while dataset-dependent, precise tuning is not overly critical. These results demonstrate that AdaTS's complementary components effectively adapt to diverse temporal characteristics, enabling robust representation learning across different contexts.

**Computational Overhead.** We evaluate the computational overhead of AdaTS by comparing it with the vanilla CL counterparts. Table 2 shows the model sizes, pretraining overhead, and inference time for each model. The results demonstrate that AdaTS introduces minimal computational overhead (0.26-0.36% in size, 5.19-5.60% in pretraining time) while significantly improving performance. Moreover, since soft assignments are only performed during pretraining, AdaTS does not introduce additional inference time overhead, making it suitable for applications requiring low-latency inference.

Table 3 compares the time and memory overhead of different distance functions, including Euclidean, Cosine, DTW, and TFC on ACIDS dataset. Results show that AdaTS's TFC similarity metric maintains comparable computational overhead to simple metrics like Euclidean and Cosine similarity while providing better performance through frequency-domain analysis. Additionally, compared to GPU-optimized DTW, TFC achieves over 40x speedup in pretraining processing time and 2x reduction in memory overhead, particularly beneficial for high-bandwidth data. Furthermore, systematic scalability experiments across varying sequence lengths (2s-16s) demonstrate that AdaTS maintains consistent 3-7% overhead with linear scaling characteristics (detailed analysis in Appendix D.2).

## 3.3 Ablation Studies

This section evaluates AdaTS's key components through ablation studies on ACIDS and PAMAP2 datasets, using both TS2Vec and FOCAL as backbones.

**Component Analysis.** Table 4 analyzes AdaTS's core components by comparing five configurations: (i) **Baseline** (base model: TS2Vec or FOCAL); (ii) **noOrd** (adds Dynamic Temporal Assignment (DTA), excludes Ordinal Consistency (OC)); (iii) **noTemp** (adds OC, excludes Temporal Assignment (TA)); (iv) **wStaticTemp** (OC + Static TA (STA), akin to SoftCLT); and (v) **AdaTS** (full: OC + DTA). Components are added to the baseline. Results show that both OC (seen in noTemp vs. Baseline) and TA (seen in noOrd vs. Baseline) individually contribute to performance improvements across both backbones. Comparing wStaticTemp (OC + STA, akin to SoftCLT's temporal mechanism) with the full AdaTS (OC + DTA) highlights the additional benefit of our dynamic temporal assignment, particularly on the more non-stationary ACIDS dataset. Ordinal consistency (noTemp) tends to yield larger gains on PAMAP2, with its lower frequency characteristics, by effectively capturing relative physical similarities in different activities. In contrast, dynamic temporal assignment (comparing AdaTS to wStaticTemp) shows larger gains on ACIDS, which exhibits higher dynamics diversity and frequency complexity. This demonstrates the complementary nature of AdaTS's components, enabling it to effectively handle diverse temporal characteristics and enhance models incorporating both simple (e.g., TS2Vec) and complex (e.g., FOCAL) temporal constraints.

**Effect of Similarity Functions.** Table 5 compares the effectiveness of different similarity metrics $D$ in Algorithms 1 and 3 for capturing temporal relationships. On PAMAP2, Euclidean distance performs better than DTW and Cosine similarity, demonstrating that simple point-wise metrics can be effective for lower-frequency signals with more regular temporal patterns. On the other hand, Cosine similarity shows better performance on ACIDS due to its robustness to amplitude variations in signals with higher dynamics. TFC outperforms these conventional metrics on both datasets through its generalizable and robust frequency-domain analysis. To validate TFC's design choice of STFT over alternative time-frequency transforms, we conducted systematic comparisons with wavelet transforms (Appendix E). Results demonstrate TFC's superior generalizability across diverse datasets (+2.10% accuracy on PAMAP2, +1.94% on MOD), while wavelets show specialized advantages only in highly transient scenarios.

## 4 Related Work

**Self-Supervised Learning (SSL).** SSL generally categorizes into contrastive learning (CL) [7, 23, 8, 5, 32, 36], masked autoencoders (MAE) [25, 21, 59, 28], and others. CL, which maximizes similarity of representations of positive pairs while minimizing that of negative pairs, has been widely adopted for its transferability and generalizability. Apart from treating augmented views as positive pairs (similar) in the unimodal context, CL has also been extensively applied to multimodal data, considering samples from different modalities as positive pairs [54, 37, 44, 45, 73].

**Contrastive Learning in Time Series.** For TS analysis, various designs of augmentations [64] and contrasting strategies [16, 17, 66, 67, 68, 71, 55, 9, 61] have been proposed to achieve better performance with CL, based on temporal correspondences of TS data. For physical sensing, CL has shown significant advancement using multimodal sensor data from wearable devices [43, 26, 13]. Most CL methods for TS analysis assign the same weight to all negative pairs and compute a hard contrastive loss.

**Soft Contrastive Learning.** CL typically discriminates each instance from others and treats all negative pairs equally. That often overlooks the properties of instances and pushes similar instances farther in the embedding space. Consequently, multiple approaches have been proposed to employ soft inter-sample relations [18, 20, 60]. In the TS domain, CL also faces significant challenges due to the complex nature of temporal correlations and non-stationary signal characteristics. Recent works like StatioCL [63] and SoftCLT [34] have attempted to address these limitations through non-stationarity-aware and soft similarity measures, but their reliance on sequence-level processing and caching mechanisms limits scalability and granularity. In contrast, our approach introduces efficient and adaptive similarity measures that dynamically respond to both physical and temporal characteristics of the data, enabling more robust representation learning at scale.

## 5 Discussion and Conclusions

In this paper, we introduced AdaTS, an adaptive strategy for time-series contrastive learning that effectively captures physical correlations and non-stationary characteristics in time series. By leveraging Time-Frequency Coherence as a physics-guided similarity measure to enhance ordinal consistency loss and dynamic temporal assignments, AdaTS achieves up to a 13.7% accuracy improvement across multiple datasets while maintaining computational efficiency. Its pluggable design enables seamless integration with existing contrastive learning methods, enhancing robustness, particularly in limited-label scenarios. Our extensive evaluation further highlights AdaTS's adaptability to varying sampling rates and diverse IoT applications with dynamic signal patterns. Future work includes addressing the "oversampling" problem in ordinal consistency loss, where frequent sampling of similar instances may shrink embedding space distances. In the Appendix F, we provide additional pointers and discuss the limitations of AdaTS.

## Acknowledgements

This work was partially supported by NSF CNS 20-38817, the JUMP2 SRC program, and the Boeing Company.

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

# Appendix

The appendix of this paper is structured as follows.

- Section A introduces the evaluated datasets with detailed specifications and preprocessing details.
- Section B details the compared baselines in our experiments.
- Section C introduces the used backbone model structure and their related configurations.
- Section D presents more comprehensive evaluation results including oversampling robustness experiments and computational scalability analysis.
- Section E provides systematic comparison between wavelet transforms and STFT-based TFC, demonstrating complementary strengths.
- Section F discusses limitations including STFT assumptions and extreme non-stationarity, along with future work directions.
- Section G describes the training configurations and experimental details.
- Section H discusses the broader impacts.

## A  Datasets

In this section, we provide detailed descriptions and specifications of each dataset. Table 6 presents comprehensive statistics including modalities, sampling frequencies, and segmentation details.

### A.1  Dataset Specifications

Table 6: Detailed Dataset Specifications and Preprocessing

| Dataset | Modalities (Freq) | Sample Length | Overlap | #Labels | Downstream Task |
|---|---|---|---|---|---|
| ACIDS [41] | acoustic, seismic (1025Hz) | 1 sec | 50% | 27,595 | Vehicle Classification (9) |
| MOD [37] | acoustic (8kHz), seismic (100Hz) | 2 sec | 0% | 7,335 | Multi-task: Vehicle (7), Distance (4), Speed (4) |
| PAMAP2 [48] | acc, gyro, mag, light (50Hz) | 5 sec | 50% | 9,611 | Activity Classification (18) |
| RWHAR [53] | acc, gyro, mag (100Hz) | 2 sec | 50% | 12,887 | Activity Classification (8) |

**Segmentation Approach.** We segment raw time-series data into fixed-length samples using sliding windows. Window sizes and overlap ratios are chosen based on each dataset's temporal characteristics to capture meaningful patterns while maintaining sufficient training samples. Each segment serves as an independent sample for both pretraining and downstream classification tasks.

**UCR/UEA Benchmark Archives.** We additionally evaluate on standard time series classification benchmarks:

- **UCR Archive** (univariate): 125 datasets, 166,500+ total samples, with average sequence lengths ranging from 50 to 2,844, covering medical, motion, and sensor domains.
- **UEA Archive** (multivariate): 29 datasets, 47,000+ total samples, with average sequence lengths ranging from 8 to 5,730, covering HAR, medical, and industrial domains.

These archives exhibit more stationary patterns compared to our primary IoT datasets, yet AdaTS demonstrates consistent improvements (Figure 2b in the main paper), validating robustness across diverse temporal dynamics.

**PAMAP2** [48] comprises IMU recordings from 9 participants performing 18 diverse physical activities, including outdoor sports, household tasks, and locomotion exercises. Sensors were positioned on three body locations (chest, dominant wrist, and dominant ankle), each recording three-axis accelerometer, gyroscope, and magnetometer readings at 100 Hz. Our experiments utilize data exclusively from the wrist-mounted sensor. We employ a leave-one-out evaluation strategy, allocating seven randomly selected participants for training and two for testing, enabling cross-subject generalization assessment.

**RealWorld-HAR (RWHAR)** [53] captures multimodal sensor data from 15 participants executing 8 everyday physical activities spanning locomotion, postural transitions, and stationary positions. The dataset encompasses accelerometer, gyroscope, magnetometer, and light sensor measurements recorded at 100 Hz from waist-mounted devices. Our evaluation focuses exclusively on this waist sensor data to maintain consistency with practical deployment scenarios. Following a leave-one-out protocol, we partition the dataset using ten randomly selected subjects for training, two for validation, and three for testing.

**ACIDS** [41, 10] (Acoustic-Seismic IDentification System) represents a comprehensive vehicle classification benchmark comprising over 270 data collection runs from 9 distinct ground vehicle types operating across 3 environmental conditions developed by the U.S. Army Research Lab. Data acquisition employed two co-located acoustic and seismic sensor systems that captured vehicles traversing at constant speeds from approach through closest point of approach (CPA) to departure. Acoustic signals were band-pass filtered (25-400 Hz) and digitized at 1025 Hz using 16-bit A/D conversion. Collection parameters varied substantially: CPA distances ranged from 25 m to 100 m, while vehicle speeds spanned 5-40 km/h depending on vehicle type and terrain. This variability introduces significant domain shift challenges between training and testing partitions. We employ an 8:1:1 random split at the run level for training, validation, and testing respectively.

**MOD** [37] (Moving Object Detection) is a multimodal vibration dataset designed for moving target detection and classification. Data collection utilized sensor nodes integrating RaspberryShake 4D geophones and microphone arrays deployed at two distinct sites: a repurposed research facility and a parking environment. The geophone technology demonstrates superior sensitivity to nearby vibrations compared to conventional smartphone accelerometers. Seven target classes were captured spanning pedestrian and vehicular categories with diverse propulsion systems (electric, combustion engine, off-road variants). Each target traversed the vicinity of sensor nodes at varying speeds while sensors captured synchronized seismic (100 Hz) and acoustic (16 kHz, downsampled to 8 kHz) signals. Collection sessions lasted 40 to 60 minutes per target type. The dataset exhibits challenging real-world variability in target speed, distance, and environmental conditions. We partition samples randomly at the sample level with an 8:1:1 ratio for training, validation, and testing.

# B    Baselines

Below, we outline the baselines used in our evaluation.

**CMC** [54]: CMC is a contrastive learning framework that leverages multiview data by treating different modalities as distinct views. It minimizes the geometric distance between representations of the same sample across modalities while maximizing the distance between representations of different samples.

**Cosmo** [43]: Cosmo generates multimodal time-series representations using a contrastive fusion mechanism that maps modality embeddings onto a hypersphere. Similar features are treated as positive pairs, while dissimilar features are considered negative, facilitating cross-modal alignment.

**SimCLR** [7]: SimCLR contrasts samples across different augmented views. For each sample in the batch, random augmentations are applied to generate two unique views, which are treated as positive pairs. SimCLR brings representations of the same sample closer in the feature space while pushing apart the representations of other samples in the same batch to learn discriminative features.

**MAE** [25]: Masked Autoencoder (MAE) uses an encoder-decoder architecture to learn representations from masked reconstruction. During pretraining, a significant portion (e.g., 75%) of the input from each modality is masked. Then, separate encoders for each modality are used to extract multimodal features. After encoding, the features are fused via MLP layers, and the fused embeddings are decoded back to spectrogram inputs for reconstruction.

**TNC** [55]: TNC contrasts samples within the temporal neighborhood against temporally distant samples. It treats temporally close samples (neighbors) as positive pairs and other samples in the batch as negative pairs, with a discriminator that predicts the sample's probability of being a neighbor.

**LIMU-BERT** [65]: LIMU-BERT extends BERT's self-supervised learning approach to unlabeled IMU data, tailoring it to sensor-specific temporal patterns. Custom modifications enable it to effectively capture sequential information unique to IMU signals.

**AudioMAE** [27]: AudioMAE builds upon the MAE framework [25], leveraging a Transformer with both global and local attention mechanisms for audio representation. Audio is segmented into spectrogram patches, with a portion masked to enable efficient encoding. The model incorporates learnable embeddings for masked patches and a decoder with localized attention for spectrogram reconstruction.

**CAV-MAE** [22]: CAV-MAE combines the principles of MAE and contrastive learning to process audio-visual data. It employs separate encoders for each modality alongside a joint encoder for cross-modal feature fusion. Multi-stream processing enhances input reconstruction and cross-modal learning.

**TS2VEC** [68]: TS2VEC enhances time-series representations through contrastive objectives across hierarchical window sizes. Positive pairs are formed by augmenting the same sample and its context, while negative pairs are derived from different samples or sequences.

**TS-TCC** [16]: TS-TCC captures time-series representations by combining cross-view predictions with temporal and contextual contrastive learning. The framework generates two augmented views of each sample and extracts context vectors using an autoregressive model. These context vectors predict future timestamps in the alternate view to learn temporal alignment and context awareness.

**FOCAL** [37]: FOCAL is a contrastive learning framework specifically designed for multimodal time-series data. FOCAL factorizes the representation into shared and private subspaces to incorporate modality-common and modality-unique representations. The subspaces are enforced to be independent with geometric orthogonality constraints. To capture temporal correspondence, a temporal ranking constraint enforces temporal locality.

**Time-Series Transformer (TST)** [69]: A Transformer-based framework for multivariate time series representation learning. It uses a standard Transformer encoder architecture and is trained with a self-supervised masked reconstruction task.

**Swin Transformer** [39]: Originally proposed for computer vision, this hierarchical Vision Transformer builds representations by starting with small patches and gradually merging them. We adapt it for time-series spectrograms, using its self-supervised pre-training capabilities.

## C  Backbone Encoder

We implement the SWIN-Transformer (SW-T), a Vision Transformer (ViT) variant [39], as the primary backbone encoder for AdaTS. We adapt it specifically for processing time-frequency spectrograms derived from time-series data. Input spectrograms, potentially from multiple sensing modalities, are first segmented into non-overlapping patches. Each patch is then linearly embedded to form a sequence of embedding vectors. SW-T processes these patch embeddings through multiple stages. Each stage consists of SW-T blocks that apply self-attention within local, non-overlapping windows. To enable cross-window connections, a shifted window mechanism is employed in successive blocks. As the network depth increases, patches are progressively merged, reducing the spatial resolution while increasing the feature dimension, allowing the model to learn hierarchical representations. This design, compared to the original ViT, enhances computational efficiency while capturing both local and global contextual information from the spectrograms. For multi-modal datasets, a separate SW-T encoder can process the spectrogram of each modality independently. The AdaTS framework, including its TFC similarity metric and ordinal consistency loss, typically operates on these modality-specific feature representations before any late-stage fusion.

## D  Additional Evaluation Results

In this section, we provide additional evaluation results that are not included in the main paper.

### D.1  Representation Visualization

To qualitatively assess the quality of the learned representations, we employ t-SNE [56] to visualize the embeddings generated by the FOCAL backbone and FOCAL enhanced with AdaTS. Figure 6 presents these visualizations for the ACIDS, MOD, PAMAP2, and RWHAR datasets. In these plots, different colors correspond to distinct ground-truth labels.

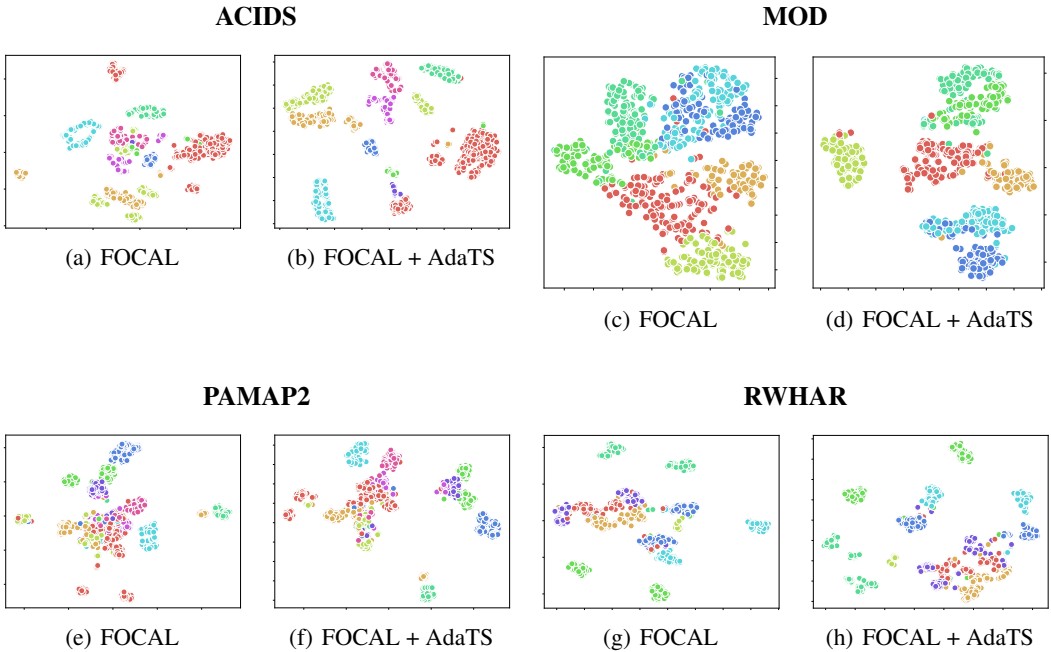

Figure 6: t-SNE visualization of embeddings on different datasets. For each dataset (**ACIDS, MOD, PAMAP2, RWHAR**), comparisons are shown for FOCAL (left subfigure of pair) and FOCAL + AdaTS (right subfigure of pair). Different colors represent different ground-truth labels. AdaTS-enhanced models generally lead to better separated and more cohesive clusters. Best viewed in color.

The visualizations generally indicate that integrating AdaTS with FOCAL leads to more distinct and well-separated clusters, suggesting an improved ability to capture the underlying data structure. For instance, on the ACIDS and MOD datasets, which are characterized by complex, high-frequency dynamics, the embeddings from FOCAL + AdaTS (Figures 6b and 6d) exhibit clearer separation between classes compared to the baseline FOCAL embeddings (Figures 6a and 6c). This improvement highlights the efficacy of AdaTS's Time-Frequency Coherence (TFC) similarity measure and dynamic temporal assignment for disentangling complex temporal patterns.

On the RWHAR dataset (Figures 6g and 6h), AdaTS further refines the representations, resulting in tighter clusters for the different human activities compared to FOCAL alone. This can be attributed to the ordinal consistency loss, which helps preserve fine-grained relative similarities between instances. The PAMAP2 dataset (Figures 6e and 6f), with its larger number of activity classes and inherent inter-class similarities, presents a more challenging scenario. While some overlap between classes persists even with AdaTS, there is a noticeable improvement in the cohesion of individual clusters and a reduction in overlap for several classes, indicating that AdaTS's components contribute positively even in such complex settings.

Overall, these t-SNE visualizations qualitatively confirm that AdaTS enhances the representational power of the FOCAL backbone, yielding feature spaces that better reflect the intrinsic structure of time-series data across diverse application domains.

### D.2 Computational Scalability Across Sequence Lengths

To comprehensively evaluate AdaTS's computational scalability, we conducted systematic experiments measuring sequence processing times across varying lengths (2s, 4s, 8s, 16s) for three different SSL frameworks. Table 7 presents the detailed results.

The results demonstrate that AdaTS maintains consistent overhead between 3% and 7% across all sequence lengths and frameworks, providing strong evidence of excellent scalability. Importantly, the overhead remains relatively stable even as sequence length increases by $8\times$ (from 2s to 16s), indicating that AdaTS's computational complexity scales linearly with the base framework without introducing quadratic or higher-order growth. This linear scaling behavior confirms that the $O(T)$ complexity of our $\kappa_n$ statistic computation (described in Section 2.4) and the efficient FFT-based

Table 7: Sequence processing time (ms) and AdaTS overhead across varying sequence lengths. AdaTS maintains consistent 3-7% overhead demonstrating linear scalability.

| Framework | Metric | 2s | 4s | 8s | 16s |
|---|---|---|---|---|---|
| FOCAL | Base Time (ms) | 2.60 | 5.13 | 10.40 | 20.37 |
| | +AdaTS Overhead | +0.16 (6.2%) | +0.31 (6.0%) | +0.46 (4.4%) | +0.96 (4.7%) |
| TS2Vec | Base Time (ms) | 2.47 | 5.07 | 9.89 | 20.83 |
| | +AdaTS Overhead | +0.11 (4.5%) | +0.21 (4.1%) | +0.43 (4.3%) | +0.76 (3.6%) |
| TS-TCC | Base Time (ms) | 2.60 | 5.44 | 11.44 | 24.52 |
| | +AdaTS Overhead | +0.09 (3.5%) | +0.24 (4.4%) | +0.49 (4.3%) | +0.82 (3.3%) |

Table 8: Noise robustness under Gaussian corruption ($\sigma$ = noise std.). AdaTS consistently outperforms FOCAL baseline.

| Dataset | Method | $\sigma = 0.1$ | | $\sigma = 0.2$ | | $\sigma = 0.5$ | | $\sigma = 1.0$ | |
|---|---|---|---|---|---|---|---|---|---|
| | | Acc | F1 | Acc | F1 | Acc | F1 | Acc | F1 |
| PAMAP2 | FOCAL | 0.8425 | 0.8302 | 0.8494 | 0.8370 | 0.8562 | 0.8464 | 0.8557 | 0.8448 |
| | +AdaTS | **0.8596** | **0.8488** | **0.8584** | **0.8493** | **0.8612** | **0.8508** | **0.8648** | **0.8587** |
| ACIDS | FOCAL | 0.9338 | 0.7947 | 0.9429 | 0.8199 | 0.9365 | 0.7809 | 0.9146 | 0.7859 |
| | +AdaTS | **0.9479** | **0.8629** | **0.9520** | **0.8375** | **0.9384** | **0.8456** | **0.9374** | **0.8590** |

TFC operations (Section 2.2) enable practical deployment across varying sequence lengths and temporal resolutions. The slight variation in overhead percentages across frameworks (3.3-6.2%) reflects differences in their baseline computational characteristics, but all remain within a narrow and acceptable range for real-world applications.

### D.3 Noise Robustness Analysis

To evaluate AdaTS's robustness under noisy conditions, we conducted systematic experiments with Gaussian noise corruption at multiple levels ($\sigma = 0.1, 0.2, 0.5, 1.0$ relative to signal standard deviation) on the PAMAP2 and ACIDS datasets.

Table 8 presents comprehensive results comparing the FOCAL baseline with FOCAL + AdaTS across different noise levels. FOCAL + AdaTS consistently outperforms the baseline across all noise levels, with F1 improvements up to +7.31% on ACIDS under extreme corruption ($\sigma = 1.0$). While FOCAL's frequency-domain processing provides baseline robustness on PAMAP2 (where Gaussian noise can act as regularization for low-dynamic activities), AdaTS's dynamic temporal supervision becomes crucial for high-frequency ACIDS data.

TFC's spectral coherence analysis effectively filters noise while preserving meaningful signal correlations. The frequency-domain approach inherently provides robustness by focusing on spectral patterns rather than point-wise amplitude values that are more susceptible to additive noise. Furthermore, ordinal consistency maintains relative similarity relationships when absolute metrics fail under extreme noise corruption. By preserving the ordering of instance similarities rather than exact distances, OCL ensures that the learned representations remain semantically meaningful even when individual similarity measurements become unreliable due to noise.

These results demonstrate that AdaTS's design choices—frequency-domain analysis through TFC and ordinal consistency learning—provide inherent robustness to environmental noise and signal corruption, making it suitable for real-world deployments where sensor data quality varies significantly.

### D.4 Robustness to Extreme Class Imbalance

To evaluate AdaTS's robustness under extreme class imbalance scenarios, we conducted controlled oversampling experiments on the PAMAP2 dataset. We systematically oversampled the three most frequent activity classes (walking, ironing, lying) at ratios of $1\times$, $2\times$, $5\times$, and $10\times$, resulting in overall dataset size increases from baseline to +315%.

Table 9 presents the results of these experiments. We observe that extreme oversampling ($5\times+$) causes performance degradation in both baseline FOCAL and FOCAL + AdaTS due to embedding space shrinkage, where over-representation of similar instances collapses the embedding space. Specifically,

Table 9: Robustness to extreme class imbalance on PAMAP2. Distance regularization (DistReg) effectively prevents embedding collapse under extreme oversampling.

| Method | 1× (Baseline) | | 2× (+35%) | | 5× (+140%) | | 10× (+315%) | |
|---|---|---|---|---|---|---|---|---|
| | Acc | F1 | Acc | F1 | Acc | F1 | Acc | F1 |
| FOCAL | 0.8427 | 0.8258 | 0.8378 | 0.8318 | 0.8200 | 0.8154 | 0.8063 | 0.8014 |
| +AdaTS | 0.8637 | 0.8535 | 0.8422 | 0.8331 | 0.8154 | 0.8039 | 0.8016 | 0.7985 |
| +AdaTS + DistReg | **0.8639** | **0.8510** | **0.8504** | **0.8424** | **0.8385** | **0.8362** | **0.8339** | **0.8241** |

at 10× oversampling, vanilla AdaTS accuracy drops from 0.8637 to 0.8016 (-6.21%), while baseline FOCAL drops from 0.8427 to 0.8063 (-3.64%).

To mitigate this issue, we apply cosine-based distance regularization following the VICReg anti-collapse framework [4]. The results demonstrate that FOCAL + AdaTS + DistReg maintains robust performance even under extreme imbalance: at 10× oversampling, it achieves 0.8339 accuracy, recovering 52% of the performance loss compared to vanilla AdaTS. This validates the effectiveness of distance regularization for preventing embedding space collapse under extreme imbalance conditions, as discussed in Section 2.3.

# E   Wavelet Transform vs. STFT-based TFC Analysis

To address potential limitations of STFT's local stationarity assumption, we conducted comprehensive experiments comparing Continuous Wavelet Transform (CWT) with Morlet wavelets as an alternative to STFT in our TFC similarity computation. This appendix presents a detailed analysis of their complementary strengths across different signal characteristics.

## E.1   Overall Performance Comparison

Table 10 presents the performance comparison between wavelet-based AdaTS and STFT-based TFC AdaTS across all primary datasets using identical experimental conditions and hyperparameters.

Table 10: Performance Comparison: Wavelet Transform vs. STFT-based TFC

| | ACIDS | | PAMAP2 | | RWHAR | | MOD | |
|---|---|---|---|---|---|---|---|---|
| Method | Acc | F1 | Acc | F1 | Acc | F1 | Acc | F1 |
| Wavelet AdaTS | 0.9525 | 0.8346 | 0.8438 | 0.8353 | **0.9533** | **0.9577** | 0.9511 | 0.9496 |
| TFC AdaTS | **0.9571** | **0.8480** | **0.8648** | **0.8560** | 0.9527 | 0.9568 | **0.9705** | **0.9700** |
| Improvement | +0.46% | +1.34% | +2.10% | +2.07% | -0.06% | -0.09% | +1.94% | +2.04% |

TFC outperforms wavelets on most datasets (ACIDS: +1.78% F1, MOD: +1.94% Acc, PAMAP2: +2.10% Acc), while wavelets show competitive performance on RWHAR. This demonstrates TFC's superior generalizability across diverse signal characteristics.

## E.2   ACIDS Case Study: Dynamic-Based Analysis

To understand the complementary strengths of both approaches, we analyzed ACIDS vehicle classification by categorizing vehicles based on movement dynamics. Table 11 presents results grouped by vehicle dynamics.

Table 11: ACIDS Vehicle Category Classification by Dynamics

| Category | Vehicle Types | Wavelet Acc | TFC Acc | Best | #Samples |
|---|---|---|---|---|---|
| Heavy vehicles | 1, 2, 8, 9 | 94.7% | **96.6%** | TFC (+1.9%) | 937 |
| Medium vehicles | 3, 5 | 81.8% | **82.7%** | TFC (+0.9%) | 313 |
| Light vehicles | 6, 7 | **91.3%** | 87.5% | Wavelet (+3.8%) | 104 |

**Key Findings:**

- **TFC advantages:** Consistent performance on heavy and medium vehicles (largest sample sizes: 937 and 313 samples) with steady, predictable motion patterns. TFC's frequency-domain coherence effectively captures consistent spectral relationships in these scenarios.

- **Wavelet advantages:** Superior performance on light vehicles (104 samples) exhibiting highly agile maneuvers and rapid acceleration changes, where wavelet's multi-resolution analysis better captures transient dynamics.

### E.3 Individual Vehicle Type Breakdown

Table 12 provides granular analysis at the individual vehicle type level, revealing systematic performance patterns.

Table 12: Individual Vehicle Type Performance Analysis

| Type | Category | Wavelet | TFC | Best | Adv. | Samples | Rationale |
|------|----------|---------|------|------|------|---------|-----------|
| Type 2 | Heavy | 92.9% | **98.5%** | TFC | +5.6% | 326 | Consistent patterns favor coherence |
| Type 9 | Heavy | 93.9% | **96.6%** | TFC | +2.7% | 179 | Low-frequency signatures |
| Type 8 | Heavy | 97.0% | **99.3%** | TFC | +2.3% | 134 | Steady, low-dynamics movement |
| Type 7 | Light | **60.9%** | 43.5% | Wavelet | +17.4% | 23 | Highly agile, rapid maneuvers |

**Analysis:** TFC demonstrates superior performance across the largest sample sizes (Types 2, 8, 9), indicating strong generalizability. The largest wavelet advantage (+17.4% for Type 7) occurs in the smallest sample size (23 samples) with the most dynamic vehicle type, suggesting a specialized rather than generalizable advantage for extreme transient scenarios.

### E.4 Implications and Recommendations

Our findings reveal systematic complementary strengths:

1. **TFC's broad applicability:** Superior performance across general cases, steady patterns, and larger datasets. Recommended as the default choice for most time-series representation learning scenarios.

2. **Wavelet's specialized strength:** Advantages in highly dynamic scenarios with abrupt changes (*e.g.,* rapid maneuvers, mechanical faults). Beneficial for applications specifically targeting transient phenomena.

3. **Unified framework potential:** AdaTS can leverage both approaches by selecting the appropriate time-frequency transform based on signal dynamics, potentially achieving best-of-both-worlds performance.

**Computational Considerations:** While wavelets offer theoretical advantages for non-stationary signals, STFT-based TFC provides superior computational efficiency through FFT optimizations, making it more practical for large-scale deployments. Future work could explore adaptive selection mechanisms that choose between STFT and wavelet transforms based on detected signal characteristics.

## F   Limitations and Future Work

### F.1   Ordinal Consistency Oversampling

One potential limitation of the proposed instance-wise ordinal contrastive loss is the "oversampling" problem: if a group of very similar or closely related instances is sampled much more frequently than others, their calculated similarity may remain extremely close, concentrating ordinal pairs and shrinking distances in the embedding space. Our experiments (Section 2.3 and the added evaluations in Appendix D) demonstrate that distance regularization following the VICReg framework [4] effectively mitigates this issue, recovering 52% of performance loss under extreme $10\times$ oversampling conditions. We recommend applying distance regularization when dataset imbalance exceeds a $5\times$ oversampling ratio.

## F.2 STFT-Based Limitations for Extreme Non-Stationarity

While our STFT-based TFC approach effectively captures time-frequency characteristics across diverse datasets, it inherently assumes local stationarity within analysis windows. This assumption may limit performance in scenarios with extreme instantaneous non-stationarity:

**Highly Transient Signals:** Signals with abrupt, discontinuous changes (*e.g.,* sudden mechanical faults, seismic events, rapid maneuvers) may not be optimally characterized by STFT's fixed-resolution time-frequency decomposition. Our wavelet analysis (Appendix E) demonstrates that CWT with Morlet wavelets can provide advantages in highly agile scenarios (Type 7 vehicles: +17.4% improvement), though TFC maintains superior generalizability across most cases.

**Extreme Instantaneous Dynamics:** The current $\kappa_n$ statistic measures average adjacent dissimilarity, which may not fully capture instantaneous step changes or sudden environmental shifts. While effective for gradual non-stationarity (as demonstrated on ACIDS and MOD datasets), extreme instantaneous changes represent an edge case requiring specialized treatment.

## F.3 Future Directions

**Adaptive Time-Frequency Representations:** Future work will explore advanced representations like wavelet transforms, Wigner-Ville distributions [57], or adaptive selection mechanisms that choose between STFT and wavelets based on detected signal characteristics. This could enable AdaTS to leverage the complementary strengths of different time-frequency decompositions.

**Enhanced Temporal Statistics:** Develop more adaptive and dynamic statistics derived from temporal similarities. Incorporating real-time anomaly detection or event-driven weighting could enhance adaptability to sudden signal changes, particularly beneficial for streaming data applications where non-stationarity patterns evolve unpredictably.

**Streaming and Online Adaptation:** Extend AdaTS for real-time streaming scenarios with incremental $\kappa_n$ computation (O(1) complexity per sample as discussed in the rebuttal). This would enable edge IoT deployment for responsive applications like activity recognition transitions or predictive machinery maintenance.

# G   Training Configurations and Experiment Details

This section details the training strategies, hyperparameter settings, and implementation specifics for pretraining and fine-tuning models with AdaTS. The primary configurations are summarized in Table 13, and SW-T model-specific parameters are in Table 14.

Table 13: General Training Configurations for AdaTS.

| Parameter | Pretraining | Fine-tuning |
|---|---|---|
| Optimizer | AdamW | Adam |
| Weight Decay | 0.05 | 0.05 |
| Max Learning Rate | 1e-4 (Default) | 1e-2 (Default) |
| Min Learning Rate (Pretrain) | 1e-7 | N/A |
| Learning Rate Scheduler | Cosine Annealing | Step Decay |
| LR Decay Factor (Fine-tuning) | N/A | 0.2 |
| LR Decay Period (Fine-tuning) | N/A | 50 epochs |
| Warmup Epochs (Pretrain) | 10 | N/A |
| Epochs | MOD, ACIDS: 2500 RWHAR, PAMAP2: 1000 | 200 |
| Batch Size | 256 (64 sequences) | 128 |
| Sequence Length ($T_{seq}$) | 4 | N/A |
| Temperature ($\tau$) | 0.1 (Default for CL) | N/A |

**Training Strategy Details**: During pretraining, we employ the AdamW optimizer with a cosine annealing learning rate scheduler, including a brief warmup period. We randomly sample batches consisting of 64 sequences, where each sequence comprises 4 consecutive samples, totaling 256 samples per batch. The constitution of these sequences is determined at the start of each epoch. For fine-tuning, we switch to the Adam optimizer with a step learning rate decay schedule. The specific

Table 14: SW-T Model Configurations.

| Datasets | MOD | ACIDS | RealWorld-HAR | PAMAP2 |
|---|---|---|---|---|
| Dropout Ratio | 0.2 | 0.2 | 0.2 | 0.2 |
| Patch Size | aud: [1, 40], sei: [1,1] | [1, 8] | [1, 2] | [1, 2] |
| Window Size | [3, 3] | [2,4] | [3, 3] | [3, 5] |
| Block Numbers | [2, 2, 4] | [2, 2, 4] | [2, 2, 2] | [2, 2, 2] |
| Block Channels | [64, 128, 256] | [64, 128, 256] | [32, 64, 128] | [32, 64, 128] |
| Head Num | 4 | 4 | 4 | 4 |
| Mod Fusion Channel | 256 | 256 | 128 | 128 |
| Mod Fusion Head Num | 4 | 4 | 4 | 4 |
| Mod Fusion Block | 2 | 2 | 2 | 2 |
| FC Dim | 512 | 512 | 256 | 128 |

learning rates and epoch counts for pretraining vary by dataset as indicated in Table 13, chosen to ensure convergence. Hyperparameters for AdaTS's components, such as the loss weights $\lambda_{oc}$ and $\lambda_t$ (Equation 1) and the dynamic temporal assignment temperature $\tau_T$, are tuned based on performance on a held-out validation set for the primary downstream task (e.g., classification accuracy). For UCR/UEA datasets, we adhere to the implementation and training configurations of TS2Vec [68] and SoftCLT [34] when integrating AdaTS.

**Implementation & Computation**: We develop the code from the open-source implementations of foundational models and techniques [7, 37, 39, 34] using PyTorch 2.0.1. We will release our code upon acceptance. Evaluations are performed on NVIDIA RTX 6000 Ada GPUs with 48GB of memory. The training time varies from a few minutes for fine-tuning to approximately one day for pretraining on the largest datasets with the SW-T backbone.

# H   Broader Impacts

This paper proposes AdaTS, an adaptive time-series representation learning framework that offers several positive societal impacts. By improving the accuracy and robustness of time-series analysis across diverse domains, AdaTS can enhance IoT applications in healthcare monitoring, industrial systems, and environmental sensing. The improved performance with limited labeled data makes machine learning more accessible for resource-constrained IoT settings where obtaining annotations is expensive or impractical. Additionally, AdaTS's compute efficiency compared to traditional time-series similarity metrics promotes more environmentally sustainable AI applications by enabling effective learning without excessive computational requirements.

However, we acknowledge several potential negative implications. Enhanced time-series analysis capabilities could facilitate more pervasive monitoring in sensitive contexts like personal activity tracking, potentially compromising user privacy. To address these concerns, future research should focus on exploring human-sensing correlations and establishing comprehensive ethical guidelines for IoT applications, particularly in domains involving human activity recognition. We acknowledge that our studies did not use real-world human subject data, and we are committed to ensuring that future research adheres to ethical standards and prioritizes user privacy.

As time-series sensing and computing become more ubiquitous, frameworks like AdaTS that efficiently extract meaningful patterns from unlabeled data will play an increasingly important role in shaping how we interact with and benefit from IoT technologies.

