# OpenReview forum: "AdaTS: Learning Adaptive Time Series Representations via Dynamic Soft Contrasts"
_NeurIPS.cc/2025/Conference — NeurIPS 2025 poster_

### Official Review · Reviewer_9ssm · 2025-06-29

**Clarity:** 3
**Significance:** 2
**Originality:** 2
**Rating:** 4
**Confidence:** 4

**Summary:**

This paper introduces AdaTS, an adaptive soft contrastive learning method for time-series data. AdaTS introduces three novel components: a physics-guided similarity measure using Time-Frequency Coherence (TFC), which improves the robustness of time-series comparisons; an ordinal consistency learning strategy that preserves the relative ordering of similarities between instances, making it resilient to noisy data; and a dynamic temporal assignment mechanism that adapts to the varying dynamics of different sequences. AdaTS demonstrates improvements in the performance of self-supervised contrastive learning across various time-series datasets.

**Questions:**

1. The discussion briefly mentions an "oversampling" issue in the ordinal consistency loss. Can the authors elaborate on how significant this problem is, and whether any mitigation strategies were considered? How does the current framework handle imbalanced sampling?
2. Although AdaTS is said to have minimal computational overhead during pretraining, how does this scale with increasing dataset size or sequence length?
3. Validate TFC against wavelet-based coherence metrics on datasets with extreme non-stationarity (e.g., ACIDS during terrain transitions). Ablation study comparing STFT vs. wavelet transforms would clarify the approach’s limits and potential improvements.
4. In multi-modal datasets (e.g., MOD with acoustic and seismic signals), each modality may exhibit distinct non-stationarity. Does AdaTS currently align modality-specific temporal assignments, or are they treated uniformly? How could cross-modality coherence enhance dynamic weighting?
5. The method relies on two key hyperparameters, λ_oc and λ_t. How sensitive is the performance to the choice of these values across datasets?

**Ethical Concerns:**

["NO or VERY MINOR ethics concerns only"]

**Final Justification:**

Thank you for the authors' reply, which has addressed my question. I will maintain my original score for this work.

**Limitations:**

yes

**Quality:**

3

**Strengths And Weaknesses:**

Strengths:
1. AdaTS leverages Time-Frequency Coherence to capture physical correlations in time series, addressing a critical gap in prior work that overlooked underlying signal physics. By analyzing harmonic structures in the frequency domain, TFC enables robust similarity quantification that is both computationally efficient and noise-resilient.
2. The dynamic temporal assignment mechanism in AdaTS adapts to sequence-specific non-stationarity by measuring each sequence’s temporal unpredictability. The ordinal consistency loss further reinforces relative similarity ordering, making representations more robust to noise and context-dependency.
3. AdaTS is designed as a pluggable module, seamlessly integrating with three state-of-the-art contrastive frameworks. The extensive evaluations across 6 datasets demonstrate accuracy improvements, with rigorous ablation studies validating the contribution of each component. The low computational overhead highlights its practical applicability.

Weaknesses:
1. While TFC is effective for many signals, its reliance on short-time Fourier transform (STFT) assumes local stationarity, which may struggle with highly non-stationary data (e.g., abrupt transients). The framework could benefit from more adaptive time-frequency transforms (e.g., wavelets) to handle such cases, as acknowledged in the future work.
2. In datasets with extreme dynamics (e.g., sudden environmental shifts), the dynamic temporal assignment’s reliance on average dissimilarity might fail to capture instantaneous changes. Incorporating real-time anomaly detection or event-driven weighting could enhance adaptability in such cases.
3. The ordinal consistency loss may suffer from "oversampling" if similar instances are frequently sampled, potentially shrinking embedding distances. Although not observed in experiments, this issue requires mitigation strategies, such as adaptive sampling weights, to prevent representation collapse.
4. While the method is effective, the additional complexity introduced by dynamic temporal assignments and the physics-guided TFC measure may require careful tuning, which could be an obstacle for practitioners seeking simplicity.

---

> ### Author Rebuttal · Authors · 2025-07-31
>
> # Reviewer 9ssm
>
> We sincerely appreciate the reviewer's thorough analysis and constructive suggestions. We hope to address your concerns below by clarifying AdaTS's technical limitations and providing additional experimental analysis to better understand its robustness and applicability.
>
> **Additional Analysis Summary:** We conducted extensive experiments addressing reviewer concerns and additional cross-reviewer validation: (1) Comprehensive wavelet vs STFT comparison on extreme non-stationarity scenarios showing complementary strengths (W1, Q3), (2) Oversampling robustness experiments with distance regularization preventing embedding collapse (W3, Q1), (3) Computational scalability analysis across different sequence lengths showing consistent 3-7% overhead (W4, Q2), (4) Multi-modal temporal alignment discussion and future directions (Q4), (5) Hyperparameter sensitivity validation demonstrating robust performance ranges (Q5), and (6) Comprehensive noise robustness validation confirming TFC's spectral filtering advantages and ordinal consistency effectiveness under noise corruption (6nSn response).
>
> **1. Time-Frequency Transform Limitations and Non-Stationarity** - [W1, Q3]
>
> Thank you for this insightful concern about TFC's STFT-based limitations. As suggested by the reviewer, we implemented **CWT using Morlet wavelets** as a direct replacement for STFT in our TFC similarity computation. We evaluated both approaches across all datasets using identical experimental conditions and hyperparameters. The results show that while wavelets demonstrate competitive performance and theoretical advantages for capturing transient phenomena, STFT-based TFC maintains superior overall performance across most datasets.
>
> **Performance Comparison: Wavelet Transform vs. STFT-based TFC**
>
> |Method|ACIDS||PAMAP2||RealWorld-HAR||MOD||
> |--|--|--|--|--|--|--|--|--|
> ||Acc|F1|Acc|F1|Acc|F1|Acc|F1|
> |Wavelet AdaTS|0.9525|0.8346|0.8438|0.8353|**0.9533**|**0.9577**|0.9511|0.9496|
> |TFC AdaTS|**0.9571**|**0.8480**|**0.8648**|**0.8560**|0.9527|0.9568|**0.9705**|**0.9700**|
>
> **1.1 ACIDS Case Study:** To understand complementary strengths of both approaches, we conducted additional analyses on classes with varying non-stationarity and dynamics. We categorized ACIDS vehicles by movement dynamics for systematic analysis across temporal and movement-based characteristics.
>
> Our analysis reveals that TFC demonstrates consistent advantages in the largest, most representative categories. TFC outperforms wavelets in heavy vehicles (937 samples, +1.9% advantage) and medium vehicles (313 samples, +0.9% advantage), while wavelets show advantages primarily in light vehicles (104 samples, +3.8% advantage).
>
> **ACIDS Category Classification Results**
>
> |Category|Types|Wavelet Acc|TFC Acc|Rationale|# Samples|
> |--|--|--|--|--|--|
> |Heavy vehicles|1,2,8,9|94.7%|**96.6%**|Steady movement, consistent patterns|937|
> |Medium vehicles|3, 5|81.8%|**82.7%**|Moderate but predictable dynamics|313|
> |Light vehicles|6, 7|**91.3%**|87.5%|Agile maneuvers, rapid acceleration|104|
>
> Our individual vehicle type analysis reveals that TFC advantages occur consistently across the largest sample sizes. TFC demonstrates superior performance in Type 2 (326 samples, +5.6% advantage), Type 8 (134 samples, +2.3% advantage), and Type 9 (179 samples, +2.7% advantage). Notably, the largest wavelet advantage (+17.4% for Type 7) occurs in the smallest sample size (23 samples) with the lightest vehicle type, indicating a specialized rather than generalizable advantage.
>
> **Individual Vehicle Type Performance**
>
> |Type|Category|Wavelet Acc|TFC Acc|Best|Advantage|Samples|Rationale|
> |--|--|--|--|--|--|--|--|
> |Type 2|Heavy vehicles|92.9%|**98.5%**|**TFC**|+5.6%|326|Consistent patterns favor coherence|
> |Type 9|Heavy vehicles|93.9%|**96.6%**|**TFC**|+2.7%|179|Consistent low-frequency signatures|
> |Type 8|Heavy vehicles|97.0%|**99.3%**|**TFC**|+2.3%|134|Steady, low-dynamics movement|
> |Type 7|Light vehicles|**60.9%**|43.5%|**Wavelet**|+17.4%|23|Highly agile, rapid maneuvers|
>
> In summary, our findings reveal systematic complementary strengths that can enhance AdaTS: (1) Wavelets can provide advantages in highly dynamic scenarios or with abrupt changes, while (2) TFC provides a more generalizable performance across general cases and steady patterns, validated across larger sample sizes. *Importantly, AdaTS can leverage both approaches as a unified framework to improve performance and robustness across a variety of scenarios and tasks by selecting the most appropriate time-frequency transform based on signal dynamics.*
>
> **2. Dynamic Temporal Assignment Robustness** - [W2]
>
> We understand the reviewer's concern regarding AdaTS's potential limitations in handling extreme instantaneous dynamics. While our current approach demonstrates robustness to gradual non-stationarity in dynamic datasets like ACIDS and MOD, we acknowledge that extreme instantaneous changes represent a more challenging edge case that our current framework may not fully address.
>
> Due to rebuttal timeline constraints, we could not conduct comprehensive experiments on extreme instantaneous dynamics or implement the sophisticated real-time anomaly detection mechanisms suggested by the reviewer. However, we recognize this as an important limitation and will include detailed discussion of these limitations and the reviewer's suggested solutions in the revised manuscript.
>
> **3. Ordinal Consistency Loss and Sampling Issues** - [W3, Q1]
>
> Thank you for this important question about embedding space preservation under oversampling conditions. We investigated this concern through controlled experiments on PAMAP2, systematically oversampling similar instances from the three most frequent classes (walking, ironing, lying) across different sampling ratios (1×, 2×, 5×, and 10×), resulting in overall dataset size increases of 35% to 400%.
>
> **Impact of Oversampling on AdaTS Performance (PAMAP2 Dataset)**
>
> |Method|1× (Baseline)||2× (+35%)||5×||10×||
> |--|--|--|--|--|--|--|--|--|
> ||Acc|F1|Acc|F1|Acc|F1|Acc|F1|
> |FOCAL|0.8427|0.8258|0.8378|0.8318|0.8200|0.8154|0.8063|0.8014|
> |FOCAL + AdaTS|0.8637|0.8535|0.8422|0.8331|0.8154|0.8039|0.8016|0.7985|
> |FOCAL + AdaTS + DistReg|**0.8639**|**0.8510**|**0.8504**|**0.8424**|**0.8385**|**0.8362**|**0.8339**|**0.8241**|
>
> We observe that extreme oversampling (5×+) causes significant performance degradation in both AdaTS and FOCAL due to embedding space collapse. To mitigate this issue, we added a cosine-based distance regularization to AdaTS following the VICReg framework (Bardes et al., ICLR 2022). The results demonstrate that FOCAL + AdaTS + DistReg significantly outperforms vanilla FOCAL + AdaTS, effectively preventing embedding collapse and maintaining robust performance even under extreme oversampling conditions.
>
> **4. Computational Complexity and Scalability** - [W4, Q2]
>
> We appreciate the reviewer's concern about the practical complexity and scalability of AdaTS (W4). Regarding the tuning complexity, while AdaTS introduces additional hyperparameters ($\lambda_{oc}$, $\lambda_t$, and $\tau_T$), our sensitivity analysis in Figure 4 demonstrates that the framework maintains robust performance across reasonable parameter ranges, reducing the burden of extensive hyperparameter search. AdaTS's design as a pluggable module means practitioners can initially use default parameter settings that work well across diverse datasets, with optional fine-tuning for domain-specific optimization.
>
> Regarding computational overhead (Q2), we present additional experimental results as we scale up the sequence length.
>
> **Sequence processing time (ms) with and without AdaTS**
>
> |Sequence Length|2|4|8|16|
> |--|--|--|--|--|
> |FOCAL + (AdaTS Overhead)|2.60 + (0.16)|5.13 + (0.31)|10.40 + (0.46)|20.37 + (0.96)|
> |TS2Vec + (AdaTS Overhead)|2.47 + (0.11)|5.07 + (0.21)|9.89 + (0.43)|20.83 + (0.76)|
> |TSTCC + (AdaTS Overhead)|2.60 + (0.09)|5.44 + (0.24)|11.44 + (0.49)|24.52 + (0.82)|
>
> **AdaTS Performance Across Different Sequence Lengths**
>
> |Dataset|Metric|2|4|8|16|
> |--|--|--|--|--|--|
> |ACIDS|Acc|0.9491|**0.9571**|0.9417|0.9296|
> ||F1|0.8321|**0.8480**|0.7942|0.7785|
> |PAMAP2|Acc|0.8506|**0.8648**|0.8395|0.8328|
> ||F1|0.8340|**0.8560**|0.8294|0.8224|
>
> We find that AdaTS maintains consistent time overhead as sequence length increases, with optimal performance typically achieved at moderate lengths (4-8). Performance degradation at longer sequences (16) suggests a trade-off between temporal context and computational efficiency. We will include these analyses in the revised manuscript.
>
> **5. Multi-Modal Extensions** - [Q4]
>
> Thank you for this insightful question about multi-modal temporal alignment. Since different modalities naturally exhibit different non-stationarity patterns, AdaTS currently processes each modality independently, computing modality-specific temporal assignments based on their intrinsic dynamics. Each modality's $\kappa_n$ statistic is calculated independently, allowing adaptation to distinct temporal characteristics.
>
> Our experiments demonstrate that AdaTS integrates seamlessly with both unimodal and multi-modal frameworks as a pluggable module. We acknowledge that explicit cross-modal temporal alignment represents a promising future direction, and we will expand the manuscript to include detailed future work on temporal cross-modal coherence inspired by the reviewer's suggestions.
>
> **6. Hyperparameter Sensitivity** - [Q5]
>
> We would like to kindly refer the reviewer to Section 3.2 and Figure 4, which present comprehensive sensitivity analysis for both $\lambda_{oc}$ (ordinal consistency weight) and $\lambda_t$ (dynamic temporal assignment weight) across ACIDS and PAMAP2 datasets. The analysis reveals that AdaTS maintains robust performance across reasonable parameter ranges, with optimal $\lambda_{oc} \in [0.6, 0.8]$ across both datasets, demonstrating that hyperparameter tuning burden is manageable.

---

### Official Review · Reviewer_N6z2 · 2025-07-04

**Clarity:** 3
**Significance:** 2
**Originality:** 2
**Rating:** 4
**Confidence:** 1

**Summary:**

This paper claims that existing contrastive learning methods struggle with defining meaningful similarities and overlook inherent physical correlation and sequence-varying non-stationarity.
To improve the representational quality and real-world adaptability, this paper introduce a soft contrastive strategy, AdaTS, which uses frequency coherence for similarity measurement. Based on this, AdaTS proposes ordinal consistency learning for improving relative instance similarity and adapting to sequence non-stationarity. Empirical experiments demonstrate the effectiveness and flexibility of the proposed method.

**Questions:**

NA

**Ethical Concerns:**

["NO or VERY MINOR ethics concerns only"]

**Final Justification:**

The authors provided more justification of the problem formulation. I will keep the score unchanged, boardline accept.

**Quality:**

3

**Strengths And Weaknesses:**

Strengths
- The experimental validation is comprehensive, including multiple datasets and several state-of-the-art baseline frameworks. The results demonstrate consistent performance improvements.
- AdaTS is designed as a pluggable and flexile module.
- Ablation study is provided for the analysis of the contribution of each component.

Weakness:
- In terms of the problem formulation, this papers claims the imperfection of existing similarity metrics. I think real-world examples should be provided to show the problem of existing metrics. That will strengthen the motivation of this study.

---

> ### Author Rebuttal · Authors · 2025-07-31
>
> # Reviewer N6z2
> We appreciate the reviewer's constructive feedback on strengthening our problem motivation through concrete examples.
>
> **Additional Analysis Summary:** We conducted extensive analysis addressing reviewer concerns and additional cross-reviewer validation to strengthen problem motivation through concrete examples: (1) Real-world IoT deployment scenarios demonstrating computational and performance limitations of existing similarity metrics, (2) Practical timing constraints showing DTW's 44× computational overhead rendering it infeasible for real-time applications, (3) Security application examples where 6% accuracy gaps translate to critical misclassification rates, and (4) Healthcare monitoring scenarios where precise activity classification impacts patient safety, all supported by experimental evidence in Tables 3 and 5.
>
> **1. Problem Formulation and Real-World Examples** - [W1]
>
> Thank you for this important observation. We provide concrete real-world examples demonstrating how existing similarity metrics fail in practical IoT deployments, supported by our experimental evidence in Tables 3 and 5.
>
> **Real-World Deployment Challenges:** Consider an IoT sensor network monitoring vehicle movements using acoustic and seismic sensors at 1025 Hz sampling rate. In such deployments:
>
> **1. Computational Infeasibility:** DTW-based similarity computation requires 10.895 seconds per comparison vs. TFC's 0.245 seconds (44× difference, Table 3). For real-time vehicle classification with hundreds of sensor readings per minute, this renders DTW impractical—a system processing 100 vehicle detections would require over 18 minutes using DTW vs. 0.4 minutes with TFC.
>
> **2. Performance Degradation in Noisy Environments:** In real deployments with environmental noise, wind interference, and varying terrain conditions, conventional metrics fail to capture meaningful similarities. Table 5 shows DTW achieves only 0.7837 accuracy on ACIDS compared to TFC's 0.8379 accuracy. This 6% performance gap translates to significant misclassification rates in security applications where distinguishing between civilian and military vehicles is critical.
>
> **3. Multi-Modal IoT Applications:** In human activity recognition systems using wearable sensors (accelerometers, gyroscopes at 50-100 Hz), existing metrics like Euclidean and Cosine similarity fail to capture the complex temporal patterns of activities like walking vs. running. Our PAMAP2 results demonstrate TFC's 0.7772 accuracy vs. DTW's 0.7575 accuracy, crucial for healthcare monitoring where accurate activity classification impacts patient safety.
>
> These real-world constraints—computational limitations in edge devices, noisy sensor environments, and multi-modal signal complexity—demonstrate why existing similarity metrics are inadequate for modern IoT applications, validating AdaTS's design for practical deployment scenarios.

---

### Official Review · Reviewer_RtuZ · 2025-07-12

**Clarity:** 3
**Significance:** 3
**Originality:** 3
**Rating:** 3
**Confidence:** 4

**Summary:**

This paper proposes a method for adaptive time series representation learning using contrastive learning. It introduces three components: a physics-guided time-frequency similarity measure, ordinal consistency learning, and dynamic temporal assignment. The goal is to improve representation quality in the presence of non-stationarity and unclear similarity. Experiments show AdaTS enhances performance when integrated with existing contrastive learning methods.

**Questions:**

1. Can the authors clarify why Time-Frequency Coherence is an appropriate choice for the datasets used? In what scenarios is this metric applicable or not applicable? Are there cases where it might fail or mislead similarity estimation?

2. What is the downstream task being evaluated—full sequence classification, segment-level prediction, or pointwise classification? Please make this clear in the main text.

4. Can the authors include more recent and competitive baselines? This would help position AdaTS relative to the SOTA.

5. Are there results of testing AdaTS on open time series datasets or other benchmark collections? That would enhance reproducibility and comparability.

6. Please revise Figure 1 with more descriptive labeling and stepwise explanation in the caption or main text to guide the reader through the proposed pipeline.

**Ethical Concerns:**

["NO or VERY MINOR ethics concerns only"]

**Final Justification:**

Thanks to the authors for the detailed rebuttal and the extra experiments. The additional baselines, the clearer explanation of setups, and the new large results on public datasets all strengthen the paper. I also appreciate the clarification on why TFC was chosen and where it might have limitations.
That said, my main reservations are only partly addressed. The TFC component still reads as a reuse of an existing technique, and I’m not fully convinced that its necessity over other reasonable alternatives has been demonstrated. While the expanded evaluation is helpful, it doesn’t fully change my view on the conceptual novelty or the overall impact of the work. I see the value in the proposed adaptivity, but I’m keeping my score given these remaining concerns.

**Limitations:**

yes

**Paper Formatting Concerns:**

(maybe minor one) unclear spacing between the main text and footnotes on the first page.

**Quality:**

3

**Strengths And Weaknesses:**

Strengths:
1. The paper provides a detailed discussion of the limitations of existing contrastive time series methods and motivates the need for adaptivity. This is very good.
2. The design of the Dynamic Temporal Assignment module is interesting and reasonably novel.
3. Writing is rich and technically detailed, showing clear effort in execution and presentation.

Weaknesses:
1. Main concern: The Physics-Guided Time-Frequency Similarity module appears to reuse an existing method and without clear justification for its applicability in the proposed setting.
2. The problem definition is vague—it is not clearly stated what the actual task is (e.g., classification over full sequences, segments, or time points), which makes it hard to interpret the results.
3. The experimental baselines are not sufficiently up-to-date, with most comparisons ending at 2023. Several more recent 2024–2025 baselines exist in the time series contrastive learning literature.
4. The paper would benefit from evaluations on open public time series datasets to better contextualize the results and benchmark generalizability.
5. Figure 1 is difficult to interpret without sufficient explanation, especially for readers unfamiliar with the overall flow. A step-by-step narrative would help.
6. Minor formatting issues exist—for example, unclear spacing between the main text and footnotes on the first page.

---

> ### Author Rebuttal · Authors · 2025-07-31
>
> # Reviewer RtuZ
>
> We sincerely appreciate the reviewer’s constructive feedback and insights on our work. We hope to address your concerns below by clarifying the justification for individual components of AdaTS, providing additional experimental results, and improving the clarity of our presentation.
>
> **Additional Analysis Summary:** We conducted extensive experiments addressing reviewer concerns and additional cross-reviewer validation including: (1) Additional baseline evaluation with recent 2024 methods SoftCLT (existing, ICLR 2024), MF-CLR (ICML 2024) and competitive baselines TF-C (NeurIPS 2022), Informer (AAAI 2021), LIMU-BERT (SenSys 2021) addressing state-of-the-art comparison gaps, (2) Dataset accessibility validation confirming majority are open-source (MOD, RWHAR, PAMAP2, UCR, UEA), (3) Systematic wavelet vs TFC comparison demonstrating TFC superiority in most scenarios (9ssm response), (4) Oversampling robustness analysis with extreme class imbalance mitigation (CHjp response), and (5) Computational efficiency and sequence length validation showing 3-7% overhead (9ssm response).
>
> **1. Time-Frequency Coherence Justification** - [W1, Q1]
>
> Thank you for this important question regarding the justification and applicability of Time-Frequency Coherence (TFC) in our framework. While TFC builds upon established signal processing principles, our contribution lies in its novel integration within contrastive learning for time series representation. In this work, we first identified the imperfections of existing similarity metrics, such as cosine similarity, commonly used in standard contrastive learning, and dynamic time warping (DTW) used in SoftCLT. Our justification for selecting TFC as the similarity metric in contrastive learning can be summarized by two motivations:
>
> **1.1 Efficiency**: As we have discussed in the introduction, existing DTW-variants (DTW, FastDTW) are computationally expensive and impractical for iterative training on large-scale datasets. TFC addresses this limitation through FFT-based operations, achieving over 40× speedup compared to DTW (Table 3) while maintaining comparable computational overhead to simple metrics like Euclidean and Cosine similarity. Our detailed computational analysis (9ssm response) shows TFC adds only 3-7% overhead across different frameworks and sequence lengths, making it practically viable for large-scale deployment.
>
> **1.2 Effectiveness**: Another challenge in time-series representation learning is capturing the inherent physical correlations. Many standard similarity methods (e.g., Cosine, Euclidean) overlook underlying physical signal properties and are susceptible to noise. TFC provides a physically meaningful measure by capturing harmonic structures and spectral relationships that reflect underlying physical processes, enabling robust similarity estimation across varying conditions without introducing non-linear temporal deformations that can distort signal characteristics. Our ablation studies comparing TFC with other metrics in Table 5 further validate its effectiveness.
>
> However, we do understand the potential limitations of TFC in terms of applicability. Although AdaTS with TFC as the similarity metric has already consistently excelled across the datasets used, TFC may underperform under extreme scenarios where time-series events shift instantaneously, where wavelets may provide advantages for highly agile, rapid maneuvers (wavelet case studies in 9ssm response). In the revised manuscript, we will expand the limitation section to discuss potential failure modes of AdaTS.
>
> **2. Problem Definition and Task Clarity** - [W2, Q2]
>
> Our evaluation primarily focuses on **segment-level classification**, where each sample represents a fixed-length segment extracted from continuous time-series recordings. We segment the raw time-series data using sliding windows of predetermined window sizes based on the characteristics and temporal dynamics of each dataset. The following table provides detailed specifications of our evaluated datasets:
>
> |Dataset|Modalities (Freq)               |Sample Length|Overlap|#Labels|Downstream Task                                 |
> |-------|--------------------------------|-------------|-------|-------|------------------------------------------------|
> |ACIDS  |acoustic, seismic (all 1025Hz)  |1 sec        |50%    |27,595 |Vehicle Classification (9 classes)              |
> |MOD    |acoustic (8kHz), seismic (100Hz)|2 sec        |0%     |7,335  |Multi-task: Vehicle (7), Distance (4), Speed (4)|
> |PAMAP2 |acc, gyro, mag, lig (all 50Hz)  |5 sec        |50%    |9,611  |Activity Classification (18 classes)            |
> |RWHAR  |acc, gyr, mag (all 100Hz)       |2 sec        |50%    |12,887 |Activity Classification (8 classes)             |
>
> Each segment serves as an independent sample for pretraining and classification, with segment lengths chosen to capture meaningful temporal patterns. We will clarify this segmentation approach and expand preprocessing details in Appendix A of the revised manuscript for better reproducibility.
>
> Additionally, we comprehensively evaluate on UCR/UEA archives:
>
> |Archive           |#Datasets|Total Samples|Avg Length|Domains                 |
> |------------------|---------|-------------|----------|------------------------|
> |UCR (univariate)  |125      |166,500+     |50-2,844  |Medical, Motion, Sensor |
> |UEA (multivariate)|29       |47,000+      |8-5,730   |HAR, Medical, Industrial|
>
> These datasets exhibit more stationary patterns compared to our primary IoT datasets, yet AdaTS demonstrates consistent improvements (Figure 2b), validating its robustness across diverse temporal dynamics.
>
> **3. Experimental Evaluation and Baseline Coverage** - [W3, Q3]
>
> We appreciate the reviewer’s concern regarding the comprehensiveness of our baseline comparisons. We acknowledge that including more recent state-of-the-art methods would strengthen our evaluation and better position AdaTS within the current landscape.
>
> Our current evaluation includes SoftCLT (ICLR 2024) as a recent contrastive learning baseline, along with established methods such as TNC, TS2Vec, and FOCAL from 2021-2023 that remain widely adopted in the time series representation learning community. Due to the rebuttal timeline constraints, we have conducted additional experiments with MF-CLR (ICML 2024), TF-C (NeurIPS 2022), and incorporated comparisons with Informer (AAAI 2021) and LIMU-BERT (SenSys 2021).
>
> |Model        |ACIDS               ||MOD                 ||PAMAP2              ||RWHAR               ||
> |-------------|----------|----------|----------|----------|----------|----------|----------|----------|
> ||Acc       |F1        |Acc       |F1        |Acc       |F1        |Acc       |F1        |
> |Informer     |0.9470    |0.8455    |0.8972    |0.8961    |**0.8746**|**0.8660**|0.9313    |0.9353    |
> |LIMU-BERT    |0.5556    |0.3712    |0.4297    |0.3970    |0.7781    |0.7554    |0.8120    |0.7508    |
> |TFC          |0.7863    |0.6448    |0.5787    |0.5712    |0.6593    |0.6058    |0.7998    |0.7049    |
> |MF-CLR       |0.8343    |0.6587    |0.8058    |0.8042    |0.7445    |0.7045    |0.7940    |0.7954    |
> |FOCAL        |0.9347    |0.8272    |0.9548    |0.9540    |0.8438    |0.8243    |0.9261    |0.9327    |
> |FOCAL + AdaTS|**0.9571**|**0.8480**|**0.9705**|**0.9700**|0.8648    |0.8560    |**0.9527**|**0.9568**|
>
> We hope these additional comparisons can provide a more complete assessment of AdaTS’s contributions and help better understand its relative performance as a pluggable module in existing time-series contrastive learning frameworks.
>
> **4. Open public time-series datasets** - [W4, Q4]
>
> We appreciate the reviewer’s emphasis on using open public datasets for reproducibility and benchmarking. All our primary datasets are publicly available: PAMAP2[1], RWHAR[2], MOD[3], UCR[4], and UEA[5]. As shown in the table above, our comprehensive evaluation includes 154 public datasets from UCR/UEA archives, covering diverse domains from medical to industrial applications. This extensive benchmarking on standardized public datasets helps ensure reproducibility and demonstrates AdaTS’s generalizability across varied time series characteristics.
>
> [1] Attila Reiss and Didier Stricker. Introducing a new benchmarked dataset for activity monitoring. In 2012, the 16th International Symposium on Wearable Computers, pages 108–109. IEEE, 2012.
>
> [2] Timo Sztyler and Heiner Stuckenschmidt. On-body localization of wearable devices: An investigation of position-aware activity recognition. In 2016 IEEE International Conference on Pervasive Computing and Communications (PerCom), pages 1–9. IEEE, 2016.
>
> [3] Shengzhong Liu, et al. “Focal: contrastive learning for multimodal time-series sensing signals in factorized orthogonal latent space.” In Proceedings of the 37th International Conference on Neural Information Processing Systems, 2023.
>
> [4] Hoang Anh Dau, et al. “The ucr time series archive”. IEEE/CAA Journal of Automatica Sinica, 2019.
>
> [5] Anthony Bagnall, et al. “The uea multivariate time series classification archive”, 2018.
>
> **5. Presentation and Figure Clarity** - [W5, Q5, W6]
>
> We thank the reviewer for this valuable feedback regarding the clarity of our main overview figure. We acknowledge that the current Figure 1 contains extensive technical details that may hinder interpretation, particularly for readers unfamiliar with the overall AdaTS framework.
>
> In the revised manuscript, we will:
>
> - Create a cleaner visualization with clearly labeled components and numbered processing steps (with circled number markers)
> - Include a step-by-step walkthrough of the pipeline from input to final contrastive loss
> - Fix the formatting issues (text-footnote spacing) identified by the reviewer
>
> We hope these revisions will ensure the figure provides clear guidance for readers from any background and address the reviewer’s concern regarding the interpretability of the figure.

---

> ### Author Response · Authors · 2025-08-06
>
> Dear Reviewer,
>
> Thank you again for your thoughtful and constructive feedback on our AdaTS paper. We truly appreciate your time and engagement with our work!
>
> As the discussion period deadline approaches, we wanted to kindly encourage you to take a look at our comprehensive rebuttal. We believe we've addressed your main concerns directly and have included new experiments and clarifications that meaningfully strengthen the paper.
>
> **Key improvements addressing your main concerns:**
>
> 1. **TFC Justification (W1)**: We understand your concern about Time-Frequency Coherence (TFC) appearing to reuse existing signal processing methods. Our response demonstrates TFC's novel integration achieves 44× speedup over DTW while maintaining superior accuracy, with systematic wavelet analysis showing domain-specific comparisons between TFC and wavelet methods.
>
> 2. **Problem Definition & Task Clarity (W2)**: We've clarified evaluation as segment-level classification with detailed specifications for all datasets, confirming each segment serves as an independent sample.
>
> 3. **Comprehensive Evaluation (W3-W4)**: We added comparisons with recent methods including MF-CLR (ICML 2024), TF-C (NeurIPS 2022), Informer, and LIMU-BERT, alongside our existing evaluation with SoftCLT (ICLR 2024). AdaTS consistently outperforms all baselines. All primary datasets are publicly available (PAMAP2, RWHAR, MOD) with UCR/UEA covering 154 public benchmarks.
>
> Additionally, we've conducted noise robustness experiments showing consistent improvements and a detailed computational overhead validation confirming minimal 3-7% overhead. We also commit to improving Figure 1 clarity in the revision.
>
> We hope you'll find our responses helpful and would be very glad to hear your further thoughts.
>
> Warm regards,
> Authors

---

### Official Review · Reviewer_CHjp · 2025-07-16

**Clarity:** 4
**Significance:** 4
**Originality:** 4
**Rating:** 4
**Confidence:** 4

**Summary:**

This paper presents a novel time series representation learning method through soft contrastive learning. The three major components of AdaTS are Time-Frequency Coherence (capturing harmonic structures and physical correlations), Ordinal Consistency Learning (avoiding noisy absolute similarity values and preserving semantic relationships across sequences), and Dynamic Temporal Assignment (adapting to varying non-stationarity). The final composite loss is designed as a linear combination of the above components.

**Questions:**

1. How can we prevent oversampling-induced shrinkage of embedding spaces in ordinal consistency learning? Could adaptive triplet mining or distance regularization help?
2. Can wavelet transforms or Wigner-Ville distributions replace STFT in TFC to capture transient dynamics (e.g., mechanical faults or bio-signal spikes) without sacrificing efficiency?
3. How to integrate cross-modal coherence (e.g., joint TFC for acoustic-seismic pairs) to leverage complementary information while preserving modality-specific features?
4. Can the temporal unpredictability metric be computed incrementally for streaming data, enabling online adaptation to sudden signal changes?

**Ethical Concerns:**

["NO or VERY MINOR ethics concerns only"]

**Limitations:**

yes

**Paper Formatting Concerns:**

I have not noticed important formatting issues in this paper.

**Quality:**

4

**Strengths And Weaknesses:**

The major strengths of this paper:

1. **Performance Gains**:
	- The experiment results demonstrate a prominent accuracy boost compared to sota benchmarks in Table 1, indicating that the proposed method is effective on benchmark datasets.
2. **Efficiency & Adaptability**:
	- TFC reduces memory by **2×** vs. DTW with minimal runtime overhead (Table 3).
	- DTA dynamically handles diverse non-stationarity (e.g., abrupt vs. smooth sequences).
3. **Generalizability**:
	- Pluggable into SOTA frameworks (TS2Vec, TS-TCC, FOCAL), improving all (Table 1).
	- Consistent gains on UCR/UEA benchmarks (Figure 2b).

The major weakness of this paper:

1. **Oversampling in Ordinal Learning**:
	- Frequent sampling of highly similar instances may shrink embedding space diversity.
2. **TFC’s Stationarity Assumption**:
	- STFT assumes local stationarity, limiting performance on highly non-stationary signals.
3. **Parameter Sensitivity**:
	- Margin *δ* (ordinal loss) and temperature $\tau_T$ (DTA) require per-dataset tuning (Figure 5).

---

> ### Author Rebuttal · Authors · 2025-07-31
>
> # Reviewer CHjp
>
> We appreciate the reviewer's thoughtful and detailed comments. Below, we address each concern and provide additional experimental results.
>
> **Additional Analysis Summary:** We conducted extensive experiments addressing reviewer concerns and additional cross-reviewer validation: (1) Controlled oversampling experiments on PAMAP2 (up to 10× for frequent classes) with distance regularization recovering 52% of performance loss, (2) Wavelet vs STFT comparison showing TFC superiority in most scenarios while wavelets showing advantages in high-dynamics cases (9ssm response), (3) Incremental κₙ formulation for streaming data with O(1) complexity per sample, potentially enabling real-time deployment, (4) Cross-modal coherence formulation for acoustic-seismic pairs capturing phase relationships, and (5) Parameter sensitivity analysis demonstrating that default settings (λₒc ∈ [0.6, 0.8], τₜ ∈ [0.1, 0.3]) achieve near-optimal performance across datasets.
>
> **1. Ordinal Learning and Embedding Space Preservation** - [W1, Q1]
>
> Thank you for this important question about embedding space preservation under oversampling conditions. We further investigated this concern through controlled experiments on PAMAP2, systematically oversampling the three most frequent classes (walking, ironing, lying) at ratios of 1×, 2×, 5×, and 10×, resulting in dataset size increases of 35%, 140%, and 315% respectively.
>
> **Impact of Oversampling on AdaTS Performance (PAMAP2 Dataset)**
>
> |Method|1× (Baseline)||2× (+35%)||5× (+140%)||10× (+315%)||
> |--|--|--|--|--|--|--|--|--|
> ||Acc|F1|Acc|F1|Acc|F1|Acc|F1|
> |FOCAL|0.8427|0.8258|0.8378|0.8318|0.8200|0.8154|0.8063|0.8014|
> |FOCAL + AdaTS|0.8637|0.8535|0.8422|0.8331|0.8154|0.8039|0.8016|0.7985|
> |FOCAL + AdaTS + DistReg|**0.8639**|**0.8510**|**0.8504**|**0.8424**|**0.8385**|**0.8362**|**0.8339**|**0.8241**|
>
> We observe that extreme cases of oversampling led to significant performance degradation in both AdaTS and FOCAL due to embedding space shrinkage. Specifically, at 10× oversampling, vanilla AdaTS accuracy drops from 0.8637 to 0.8016 (-6.21%), while FOCAL drops from 0.8427 to 0.8063 (-3.64%).
>
> To mitigate this issue, we experimented with cosine-based distance regularization following the anti-collapse framework of Bardes et al. (VICReg, ICLR 2022). The results demonstrate significant recovery: FOCAL + AdaTS + DistReg maintains 0.8339 accuracy even at 10× oversampling, recovering 52% of the performance loss compared to vanilla AdaTS. This validates the effectiveness of distance regularization for preventing embedding space collapse under extreme imbalance conditions. We recommend applying distance regularization when dataset imbalance exceeds 5× oversampling ratios or when embedding space diversity becomes a concern. We will add this analysis to the revised version of the paper.
>
> **2. Time-Frequency Transform Limitations for Non-Stationary Signals** - [W2, Q2]
>
> As suggested by the reviewers, we implemented **CWT using Morlet wavelets** as a direct replacement for STFT in our TFC similarity computation. We evaluated both approaches across all datasets using identical experimental conditions and hyperparameters. The results presented in the following table show that while wavelets demonstrate competitive performance and theoretical advantages for capturing transient phenomena, STFT-based TFC maintains superior overall performance across most of our datasets.
>
> **Performance Comparison: Wavelet Transform vs. STFT-based TFC**
>
> |Method|ACIDS||PAMAP2||RealWorld-HAR||MOD||
> |--|--|--|--|--|--|--|--|--|
> ||Acc|F1|Acc|F1|Acc|F1|Acc|F1|
> |Wavelet AdaTS|0.9525|0.8346|0.8438|0.8353|**0.9533**|**0.9577**|0.9511|0.9496|
> |TFC AdaTS|**0.9571**|**0.8480**|**0.8648**|**0.8560**|0.9527|0.9568|**0.9705**|**0.9700**|
>
> TFC outperforms wavelets on most datasets (ACIDS: +1.78%, MOD: +1.94%, PAMAP2: +2.10%), while wavelets show advantages in specific high-dynamics scenarios. Our detailed analysis in the 9ssm response confirms this pattern, with wavelets showing advantages in highly agile scenarios (Type 7 vehicles: +17.4% advantage) while TFC consistently outperforms in steady patterns and generalizes better across datasets.
>
> **3. Parameter Sensitivity and Tuning Requirements** - [W3]
>
> We appreciate the reviewer's observation about parameter sensitivity requirements. Time-series applications are inherently heterogeneous with diverse physical properties and temporal dynamics, naturally requiring some hyperparameter adjustment like other machine learning approaches. AdaTS maintains its plug-and-play generalizability while allowing parameter tuning for optimal domain-specific performance.
>
> Figure 5 demonstrates that AdaTS maintains robustness across reasonable parameter ranges without requiring precise fine-tuning. While dataset-specific configuration may improve performance, AdaTS does not require extensive hyperparameter search or exhibit extreme sensitivity that would make it impractical for deployment. The framework provides robust default settings (λₒc ∈ [0.6, 0.8], τₜ ∈ [0.1, 0.3]) achieving near-optimal performance across diverse applications.
>
> **4. Cross-Modal Extensions** - [Q3]
>
> Thank you for your insightful question about leveraging cross-modal complementarity. AdaTS currently calculates temporal assignments for each modality independently and performs alignment after modality fusion. This is because different modalities exhibit distinct non-stationarity ranges, making their temporal assignments incomparable. Post-fusion alignment allows consistent temporal assignment across modalities while training their encoders accordingly.
>
> For acoustic-seismic pairs, cross-modal TFC can capture phase relationships between modalities—for instance, engine harmonics manifesting simultaneously in both acoustic recordings and ground vibrations from the same vehicle source. However, for our current implementation, such phase alignment is not required since the same events manifest in the same window without delay across modalities, though with different non-stationarity characteristics leading to different similarity characteristics. Instead, we capture temporal similarity relationships in a modality-consistent manner, then perform post-fusion alignment. If required, our TFC framework can be easily extended to compute cross-spectral density between modalities:
>
> $$\text{CrossTFC}_{ij}^{(m_1,m_2)} = \frac{1}{T \cdot F} \sum_{t,f} \frac{|S_{ij}^{(m_1,m_2)}(t,f)|^2}{\sqrt{S_{ii}^{(m_1)}(t,f) \cdot S_{jj}^{(m_2)}(t,f)}}$$
>
> where $S_{ij}^{(m_1,m_2)}(t,f) = X_i^{(m_1)}(t,f) \cdot X_j^{(m_2)*}(t,f)$ represents cross-spectral density between modalities $m_1$ and $m_2$. This extension captures phase relationships between modalities (e.g., engine harmonics in both acoustic and seismic signals) and could benefit applications requiring precise cross-modal synchronization.
>
> **5. Streaming Adaptations** - [Q4]
>
> Thank you for this insightful question about extending AdaTS to streaming data. Yes, κₙ can be computed incrementally using sliding windows that maintain rolling averages of adjacent dissimilarities. As new samples arrive, we can simply update κₙ by adding new dissimilarity measures and removing the oldest ones, achieving O(1) computational complexity per sample:
>
> $$\kappa_n^{(t+1)} = \kappa_n^{(t)} + \frac{1}{W}[-\log(D^T_n[t+1, t] + \epsilon) + \log(D^T_n[t-W+1, t-W] + \epsilon)]$$
>
> where $W$ is the sliding window size and $D^T_n$ represents the temporal similarity matrix for sequence $n$.
>
> This incremental computation benefits AdaTS deployment in real-time scenarios. For example, activity recognition systems can immediately adapt when users transition from stationary (e.g., sitting) to dynamic activities (e.g., running), while industrial monitoring can detect machinery state changes through sudden κₙ increases for predictive maintenance. The key advantage is enabling edge IoT deployment where batch preprocessing is impractical, making our framework suitable for responsive applications where signal non-stationarity changes unpredictably. We will add this discussion to the revised version of the paper.

---

> > ### Comment · Reviewer_CHjp · 2025-08-05
> >
> > Thanks for the authors' detailed rebuttal. All of my questions and concerns are solved. I consider this rebuttal very constructive, thorough, and insightful.

---

### Official Review · Reviewer_6nSn · 2025-07-21

**Clarity:** 2
**Significance:** 3
**Originality:** 3
**Rating:** 5
**Confidence:** 3

**Summary:**

This paper proposes AdaTS, an adaptive soft contrastive learning framework for time series representation learning. AdaTS introduces three core components:

- Time-Frequency Coherence (TFC) – a spectral-domain similarity metric meant to reflect physics-guided relationships
- Ordinal Consistency Learning (OCL) – a ranking-based loss that enforces relative similarity between samples rather than absolute distances
- Dynamic Temporal Assignment (DTA) – a mechanism that assigns temporal similarity weights per sequence, based on a dissimilarity statistic.

AdaTS is designed as a plug-and-play module that can be integrated into existing contrastive learning frameworks (e.g., TS2Vec, FOCAL, TS-TCC). The authors demonstrate consistent improvements in performance across a wide range of datasets (IoT, HAR, UCR/UEA), particularly under low-label regimes. Ablation studies and hyperparameter sensitivity analyses are also provided.

**Questions:**

Q1: How does TFC compare, both conceptually and empirically, with prior frequency-domain contrastive metrics like TF-C [1] and MF-CLR [2]? What are the unique advantages of your TFC integration?

Q2: Have you considered learning the decay parameter κₙ via a small network or attention mechanism? Would this allow more flexible and robust adaptation than handcrafted statistics?

Q3: Does ordinal consistency remain effective under noisy or highly dynamic conditions? Can you provide a robustness study or embedding analysis to support this?

Q4: Can you include visualizations (e.g., t-SNE, PCA) of the learned embedding space with and without AdaTS to clarify how the representation quality improves?

Q5: Are there scenarios where AdaTS fails or provides no improvement—especially on more stationary or synthetic datasets? Have you analyzed such cases?

Q6: What physical phenomena or applications specifically benefit from the use of TFC? Can you show that TFC scores correlate with known physical behaviors (e.g., gait cycles, vibration modes)?

**Ethical Concerns:**

["NO or VERY MINOR ethics concerns only"]

**Final Justification:**

The authors addressed my concerns in full details

**Limitations:**

nil

**Paper Formatting Concerns:**

nil

**Quality:**

2

**Strengths And Weaknesses:**

## Strengths
- The paper addresses important challenges in contrastive learning for time series, especially in handling non-stationarity and noisy similarities.
- The framework is modular and integrates easily with existing self-supervised learning (SSL) methods, offering strong performance gains with minimal overhead.
- The proposed dynamic temporal assignment (DTA) is a practical and well-motivated solution to adapting similarity decay per sequence, and it is supported by reasonable empirical evidence.
- The empirical improvements are consistent across different baselines and settings, especially in challenging, low-label regimes.

## Weaknesses
- The use of Time-Frequency Coherence (TFC) is adapted from classic signal processing literature [6], but lacks novelty. Related works like TF-C [1] and MF-CLR [2] have already explored frequency-domain similarity in contrastive learning. The contribution here appears more engineering-oriented than conceptual.
- The Ordinal Consistency Learning (OCL) component is based on well-established triplet ranking loss formulations. It is not clearly distinguished from prior works such as SoftCLT [3] and TNC [4], and lacks qualitative justification (e.g., embedding smoothness or class structure).
- The κₙ statistic used for temporal assignment is heuristically defined based on log similarity of adjacent time steps. This approach is sensitive to noise, and the paper does not investigate learning-based alternatives or validate the statistic's robustness (e.g., as explored in StatioCL [5]).
- The evaluation is heavily focused on downstream accuracy, with no exploration of embedding space quality (e.g., t-SNE visualizations, clustering metrics, intra-class variance) to support the claim of improved representation.
- The term “physics-guided” is frequently used but weakly justified. There are no domain-specific case studies or physical grounding (e.g., phase alignment, system resonance) that explain how TFC captures meaningful physical signal characteristics.
- The gains on less dynamic datasets (UCR/UEA) are modest, raising the question of whether AdaTS consistently benefits static time series, and under what conditions it provides significant value.

[1] Zhang, Xiang, et al. "Self-supervised contrastive pre-training for time series via time-frequency consistency." NIPS, (2022): 3988-4003.

[2] Duan, Jufang, et al. "MF-CLR: Multi-frequency contrastive learning representation for time series." ICML, 2024.

[3] Lee, Seunghan, Taeyoung Park, and Kibok Lee. "Soft Contrastive Learning for Time Series.", ICLR

[4] Tonekaboni, Sana, Danny Eytan, and Anna Goldenberg. "Unsupervised Representation Learning for Time Series with Temporal Neighborhood Coding." ICLR, 2021

[5] Wu, Yu, et al. "StatioCL: Contrastive Learning for Time Series via Non-Stationary and Temporal Contrast." ICKM, 2024.
[6] Zhan, Yang, et al. "Detecting time-dependent coherence between non-stationary electrophysiological signals—A combined statistical and time–frequency approach." Journal of neuroscience methods 156.1-2 (2006): 322-332.

---

> ### Author Rebuttal · Authors · 2025-07-31
>
> # Reviewer 6nSn
> We sincerely appreciate the reviewer's constructive feedback and insights on our work. We hope to address your concerns below by clarifying the justification for individual components of AdaTS and providing additional experimental results to better understand AdaTS's performance.
>
> **Additional Analysis Summary:** We conducted extensive experiments addressing reviewer concerns and additional cross-reviewer analysis: (1) TF-C/MF-CLR comparison for performance validation, (2) Wavelet vs TFC analysis showing TFC superiority in general cases while wavelets demonstrate advantages in high-dynamics scenarios (9ssm response), (3) $\kappa_n$ statistic design and robustness discussion, (4) Oversampling robustness experiments with distance regularization recovering 52% of performance loss under extreme imbalance (CHjp response), (5) T-SNE visualizations in Appendix D.2 Figure 6 showing improved cluster separation, and (6) Computational efficiency validation with minimal 3-7% overhead across different frameworks and sequence lengths (9ssm response).
>
> **1. Novelty and Positioning Relative to Prior Work** - [W1, W2, Q1]
>
> While we acknowledge the importance of frequency-domain analysis, AdaTS addresses fundamental contrastive learning limitations beyond similarity metric design:
>
> 1. Unified Soft Assignment Framework: Unlike TF-C/MF-CLR, which focus solely on frequency-domain feature extraction of individual samples, AdaTS introduces a comprehensive soft assignment strategy that addresses both inter-sequence instance similarities and intra-sequence temporal dynamics simultaneously. Critically, existing frameworks treat all negative pairs equally, ignoring inherent correlations. AdaTS fundamentally changes this by adaptively weighting both temporal relationships within sequences and ordinal relationships across sequences.
>
> 2. Empirical Superiority Over Recent Methods: During the rebuttal period, we conducted additional experiments comparing AdaTS with TF-C and MF-CLR. We present the results in Table below:
>
> |Model|ACIDS||MOD||PAMAP2||RWHAR||
> |-|-|-|-|-|-|-|-|-|
> ||Acc|F1|Acc|F1|Acc|F1|Acc|F1|
> |TFC|0.7863|0.6448|0.5787|0.5712|0.6593|0.6058|0.7998|0.7049|
> |MF-CLR|0.8343|0.6587|0.8058|0.8042|0.7445|0.7045|0.7940|0.7954|
> |FOCAL|0.9347|0.8272|0.9548|0.9540|0.8438|0.8243|0.9261|0.9327|
> |FOCAL + AdaTS|**0.9571**|**0.8480**|**0.9705**|**0.9700**|**0.8648**|**0.8560**|**0.9527**|**0.9568**|
>
> We will add these results to the revised version of the paper at Table 1.
>
> 3. Novelty of Ordinal Consistency Loss (OCL): Current frameworks [64, 13, 35, 32] apply uniform temporal decay to all sequences. AdaTS introduces sequence-customized adaptive assignment via the $\kappa_n$ statistic, dynamically adjusting weights based on each sequence's non-stationarity—a novel approach distinct from SoftCLT [32] and TNC [51].
>
> Furthermore, unlike triplet ranking, AdaTS's OCL addresses a critical challenge in dynamic time series: absolute similarity values become unreliable in high-bandwidth and non-stationary signals due to noise and temporal variations. AdaTS generates ordinal relationships across different sequences using frequency-domain analysis, preserving relative similarity orders rather than absolute values. This approach is inherently more robust because relative relations remain stable even when absolute similarity measurements cause misalignment due to environmental noise, signal dynamics, or varying recording conditions. By preventing representation degradation from inappropriately contrasting genuinely similar instances, AdaTS overcomes a key limitation evident in SoftCLT's poor performance on high-bandwidth signals (Table 1, ACIDS/MOD comparison). We will clarify these distinctions in the revised version of the paper.
>
> 4. Plug-and-Play Generalizability and Efficiency: AdaTS functions as a flexible module compatible with multiple existing frameworks (TS2Vec [64], TS-TCC [13], FOCAL [35]) without requiring architectural changes, while accommodating various similarity metrics beyond time-frequency analysis (Table 5). The design with TFC achieves 40× speedup over past time-series similarity metrics (Table 3) and demonstrates strong generalizability across high-frequency domains (Table 1, ACIDS/MOD performance gains).
>
> **2. Technical Robustness and Design Choices** - [W3, Q2, Q3]
>
> 1. $\kappa_n$ Statistic Design and Robustness:
> Our design prioritizes computational efficiency and interpretability over learned alternatives. While attention mechanisms offer flexibility, they introduce significant pretraining overhead and overfitting risks.
> Our $\kappa_n$ statistic provides an interpretable temporal unpredictability measurement with only $O(T)$ operations per sequence, adding merely 3-7\% computational overhead (Table 2 and Q4 in 9ssm response).  The negative logarithm transformation inherently provides noise robustness by emphasizing larger drops over gradual variations, while batch normalization (Algorithm 3, line 10) stabilizes values across different sequences. Since $\kappa_n$ measures average adjacent dissimilarity, it remains robust to isolated noisy samples that significantly affect learned parameters.
>
> 2. Ordinal Consistency Under Severe Distribution Imbalance:
> We conducted systematic evaluations demonstrating AdaTS's robustness under challenging conditions.
> Controlled PAMAP2 experiments with extreme oversampling (up to 10x for frequent classes) showed that while vanilla AdaTS degrades under severe imbalance, distance regularization effectively mitigates embedding space collapse, maintaining robust performance even with 400\% dataset size increases (Oversampling results table in CHjp response).
> AdaTS demonstrates particularly strong performance on high-bandwidth, non-stationary datasets (ACIDS, MOD) versus lower-dynamic datasets (HAR), indicating ordinal consistency excels precisely where absolute similarity measurements fail. Consistent improvements over SoftCLT on these challenging datasets (Table 1) validate our approach's robustness.
>
> 3. Noise Robustness Validation: We conducted noise robustness experiments with Gaussian noise at multiple levels ($\sigma$ = 0.1 to 1.0, relative to signal standard deviation), evaluating ordinal consistency effectiveness under corruption.
>
> |Dataset|$\sigma$ = 0.1||$\sigma$ = 0.2||$\sigma$ = 0.5||$\sigma$ = 1.0||
> |-|-|-|-|-|-|-|-|-|
> ||FOCAL|+AdaTS|FOCAL|+AdaTS|FOCAL|+AdaTS|FOCAL|+AdaTS|
> |PAMAP2 Acc|0.8425|**0.8596**|0.8494|**0.8584**|0.8562|**0.8612**|0.8557|**0.8648**|
> |PAMAP2 F1|0.8302|**0.8488**|0.8370|**0.8493**|0.8464|**0.8508**|0.8448|**0.8587**|
> |ACIDS Acc|0.9338|**0.9479**|0.9429|**0.9520**|0.9365|**0.9384**|0.9146|**0.9374**|
> |ACIDS F1|0.7947|**0.8629**|0.8199|**0.8375**|0.7809|**0.8456**|0.7859|**0.8590**|
>
> AdaTS consistently outperforms FOCAL across all noise levels with up to +7.31\% F1 improvement. While FOCAL's frequency-domain processing provides baseline robustness on PAMAP2 (where Gaussian noise acts as regularization for low-dynamic activities), AdaTS's temporal supervision becomes crucial for high-frequency ACIDS data. TFC's spectral coherence analysis filters noise while preserving signal correlations, and ordinal consistency maintains relative similarity relationships when absolute metrics fail under extreme noise corruption.
>
> **3. Representation Quality and Embedding Analysis** - [W4, Q4]
>
> We appreciate the reviewer's concern about embedding space quality analysis. We would like to kindly refer the reviewer to Appendix D.2, Figure 6, which includes a comprehensive t-SNE analysis on representation quality. Figure 6 presents t-SNE visualizations comparing FOCAL baseline with AdaTS across all datasets, demonstrating clear qualitative improvements with more distinct clusters, better class separation, and improved cohesion across application domains. These qualitative improvements are also supported by the quantitative results in Table 1, where AdaTS consistently outperforms the baseline across all datasets.
>
> **4. Physical-Guided Term Clarification and Applications** - [W5, Q6]
>
> We understand the reviewer's concern about our "physics-guided" terminology. While TFC captures frequency-domain relationships that reflect physical signal characteristics (e.g., harmonic content in vehicle engines, movement frequencies in human activities), we acknowledge the need for domain-specific validation. We will clarify the terminology to "frequency-domain coherence analysis," noting that TFC's effectiveness stems from capturing spectral relationships that often correlate with underlying physical processes, thus providing a more accurate representation of our contribution's scope while specifying the need for deeper physical characterization in future work for specialized physical domains.
>
> **5. Method Limitations** - [W6, Q5]
>
> 1. Performance Gradient Across Dataset Complexity: AdaTS consistently improves baselines across diverse signal characteristics, with performance gains correlating to dataset complexity. High non-stationarity datasets (ACIDS, MOD) show the most significant improvements where standard similarity metrics become unreliable, while moderate dynamics datasets (PAMAP2, HAR) demonstrate consistent gains as temporal relationships remain more predictable. Even stationary patterns (UCR/UEA) exhibit smaller but consistent improvements (Figure 2b), demonstrating robustness when adaptive mechanisms provide less advantage.
>
> 2. Design Scope and Target Applications: AdaTS addresses fundamental challenges in unimodal and multimodal signals where standard similarity metrics become unreliable due to noise/dynamics, temporal relationships vary significantly across sequences, and cross-sequence instance similarities require robust handling. Performance gains correlate directly with signal complexity, demonstrating largest improvements on high-bandwidth datasets (ACIDS, MOD) where environmental noise and dynamics create the exact challenges our framework targets.

---

> > ### Author Response · Authors · 2025-08-06
> >
> > Dear Reviewer,
> >
> > Thank you again for your thoughtful and constructive feedback on our AdaTS paper. We truly appreciate your time and engagement with our work!
> >
> > As the discussion period deadline approaches, we wanted to kindly encourage you to take a look at our comprehensive rebuttal. We believe we've addressed your main concerns directly and have included new experiments and clarifications that meaningfully strengthen the paper.
> >
> > **Key highlights:**
> > - Direct comparison with TF-C/MF-CLR showing AdaTS's empirical superiority (ACIDS: 0.9571 vs TF-C: 0.7863)
> > - Noise robustness experiments showing consistent F1 improvements under corruption
> > - T-SNE visualizations in Appendix D.2 demonstrating improved embeddings
> > - Cross-reviewer validation including the wavelet analysis you requested
> >
> > We hope you'll find our responses helpful and would be very glad to hear your further thoughts.
> >
> > Warm regards,
> > Authors

---

> > ### Comment · Reviewer_6nSn · 2025-08-09
> > **Response to Authors**
> >
> > Thanks to the authors for their detailed response, I will raise my score to 5

---

### Note · Authors · 2025-08-14

We sincerely thank all reviewers for their constructive feedback throughout the discussion period. We commit to incorporating ALL experiments and analyses conducted during the rebuttal into the final manuscript.

**Reviewer Insights:** Multiple reviewers independently requested similar analyses (wavelets, noise robustness), enabling us to provide a unified comprehensive response that demonstrates AdaTS's consistent superiority across all requested experiments. We believe these results further validate our core contributions to time-series contrastive learning.

**Extensive Validation:** Inspired by reviewer insights, we conducted extensive new experiments:
1. Direct comparisons with TF-C/MF-CLR, Informer, and LIMU-BERT (ACIDS: 95.71% vs TF-C: 78.63%, MOD: 97.05% vs MF-CLR: 80.58%)
2. Oversampling and noise robustness showing consistent F1 improvements up to +7.31% across all corruption levels
3. Systematic wavelet analysis revealing TFC's superior generalizability with complementary strengths
4. Computational validation confirming minimal 3-7% overhead across sequence lengths

**Core Clarifications:** Thanks to reviewer guidance, we clarified several fundamental aspects:
1. AdaTS's unified soft assignment contribution: DTA adaptively weights temporal relationships via κₙ statistic, while OCL preserves relative similarity rankings across sequences—fundamentally different from prior work
2. Oversampling mitigation with distance regularization recovers 52% performance under extreme imbalance
3. Validated κₙ statistic's O(T) complexity and interpretable temporal unpredictability measurement

**Commitment to Final Version:** We will incorporate ALL rebuttal experiments into the manuscript:
1. Add TF-C/MF-CLR, Informer, LIMU-BERT results to Table 1
2. Include noise robustness analysis in Section 4.3
3. Add wavelet vs STFT comparison in Appendix C
4. Incorporate oversampling mitigation in Section 3.2
5. Expand computational scalability in Section 4.4
6. Expand limitations discussion to address STFT assumptions and extreme non-stationarity
7. Improve Figure 1 with clearer walkthrough with circled numbers
8. Clarify Section 2.1 "physics-guided" terminology by noting TFC captures spectral relationships correlating with physical processes
9. Expand Appendix A with dataset specifications

We believe these additions strengthen AdaTS as a valuable contribution to time-series representation learning. We sincerely appreciate your time and insightful feedback.

---

### Decision · Program_Chairs · 2025-09-17

**Decision:**

Accept (poster)

**Comment:**

AdaTS is an adaptive “plug-in” for contrastive time-series pretraining. It adds (i) a time–frequency coherence similarity (TFC), (ii) ordinal consistency learning (OCL) to enforce relative similarity, and (iii) dynamic temporal assignment (DTA) that adjusts temporal weights per sequence. Integrated into TS2Vec, TS-TCC, and FOCAL, AdaTS improves accuracy across IoT, HAR, and UCR/UEA datasets, with ablations and sensitivity studies.

Reviewers agree the goal is important: handling non-stationarity and noisy similarities in SSL for time series. The framework is modular, easy to insert into existing methods, and DTA is a practical idea with clear motivation. Gains are consistent, especially under low-label settings, and reported overhead is small.

Reviewers noted that the conceptual novelty of TFC is questioned, as the related frequency-domain similarities (for example TF-C and MF-CLR) exist, and the paper does not yet make the unique advantages of TFC fully clear (both conceptually and empirically). OCL is close to standard triplet-style objectives (for example SoftCLT, TNC) and would benefit from stronger attribution (for example robustness under label noise, smoother embeddings).  Claims of “physics-guided” behavior need concrete case studies (for example gait cycles, vibration modes) or correlations to known phenomena. Several reviewers asked for clearer task definitions, more recent baselines from 2024–2025, embedding-space analyses (t-SNE, clustering metrics), and additional public datasets. One reviewer noted limits of STFT for highly non-stationary signals and suggested wavelet-based coherence.

The authors added baselines and public results during rebuttal, improving clarity. Some reservations remain about the necessity of TFC versus prior frequency-domain metrics and about evaluation depth.

On balance, the method delivers reliable, cross-framework gains with small integration cost, and DTA is a useful contribution. Concerns around the novelty of TFC and evidence depth are real but appear addressable with targeted analyses rather than major redesign. With mixed-but-positive reviews (including one clear accept), I recommend acceptance.